# Genetically engineered mesenchymal stem cells as a nitric oxide reservoir for acute kidney injury therapy

Haoyan Huang[1,2,3†], Meng Qian[2†], Yue Liu[1†], Shang Chen[1], Huifang Li[1], Zhibo Han[4,5], Zhong-Chao Han[4,5,6], Xiang-Mei Chen[3], Qiang Zhao[2]*, Zongjin Li[1,2,3,7]*

[1]Nankai University School of Medicine, Tianjin, China; [2]The Key Laboratory of Bioactive Materials, Ministry of Education, Nankai University, the College of Life Sciences, Tianjin, China; [3]National Key Laboratory of Kidney Diseases, Chinese PLA General Hospital, Beijing, China; [4]Jiangxi Engineering Research Center for Stem Cell, Shangrao, Jiangxi, China; [5]Tianjin Key Laboratory of Engineering Technologies for Cell Pharmaceutical, National Engineering Research Center of Cell Products, AmCellGene Co., Ltd, Tianjin, China; [6]Beijing Engineering Laboratory of Perinatal Stem Cells, Beijing Institute of Health and Stem Cells, Health & Biotech Co, Beijing, China; [7]Tianjin Key Laboratory of Human Development and Reproductive Regulation, Tianjin Central Hospital of Gynecology Obstetrics, Nankai University Affiliated Hospital of Obstetrics and Gynecology, Tianjin, China

*For correspondence:
qiangzhao@nankai.edu.cn (QZ);
zongjinli@nankai.edu.cn (ZL)

†These authors contributed equally to this work

**Abstract** Nitric oxide (NO), as a gaseous therapeutic agent, shows great potential for the treatment of many kinds of diseases. Although various NO delivery systems have emerged, the immunogenicity and long-term toxicity of artificial carriers hinder the potential clinical translation of these gas therapeutics. Mesenchymal stem cells (MSCs), with the capacities of self-renewal, differentiation, and low immunogenicity, have been used as living carriers. However, MSCs as gaseous signaling molecule (GSM) carriers have not been reported. In this study, human MSCs were genetically modified to produce mutant β-galactosidase (β-GAL[H363A]). Furthermore, a new NO prodrug, 6-methyl-galactose-benzyl-oxy NONOate (MGP), was designed. MGP can enter cells and selectively trigger NO release from genetically engineered MSCs (eMSCs) in the presence of β-GAL[H363A]. Moreover, our results revealed that eMSCs can release NO when MGP is systemically administered in a mouse model of acute kidney injury (AKI), which can achieve NO release in a precise spatiotemporal manner and augment the therapeutic efficiency of MSCs. This eMSC and NO prodrug system provides a unique and tunable platform for GSM delivery and holds promise for regenerative therapy by enhancing the therapeutic efficiency of stem cells.

## Editor's evaluation

The study provides compelling evidence that treatment with the newly designed NO prodrug, MGP, selectively triggers NO release from your genetically engineered MSCs.

The significance of the study is that it provides in vivo demonstration that MSCs can release NO in a spatiotemporal manner in a mouse model of acute kidney injury thus contributing to regeneration. This constitutes a landmark finding with profound implications that are expected to have widespread influence. The work not only shows the therapeutic efficiency of MSCs, but also holds promises for regenerative therapy by enhancing the therapeutic efficiency of stem cells. Thus, it is felt that the newly generated tools will be used by many investigators thus making the findings interesting to a broad audience.

**eLife digest** Animals are made up of cells of different types, with each type of cell specializing on a specific role. But for the body to work properly, the different types of cells must be able to coordinate with each other to respond to internal and external stimuli. This can be achieved through signaling molecules, that is, molecules released by a cell that can elicit a specific response in other cells.

There are many types of different molecules, including hormones and signaling proteins. Gases can also be potent signaling molecules, participating in various biological processes. Nitric oxide (NO) is a gas signaling molecule that can freely diffuse through the membranes of cells and has roles in blood vessel constriction and other disease processes, making it a promising therapeutic agent. Unfortunately, using artificial carriers to deliver nitric oxide to the organs and tissues where it is needed can lead to issues, including immune reactions to the carrier and long-term toxicity. One way to avoid these effects is by using cells to deliver nitric oxide to the right place.

Huang, Qian, Liu et al. have used mesenchymal stem cells – which usually develop to form connective tissues such as bone and muscle – to develop a cell-based NO-delivery system. The researchers genetically modified the mesenchymal stem cells to produce a compound called β-GAL$^{H363A}$. On its own β-GAL$^{H363A}$ does not do much, but in its presence, a non-toxic, non-reactive compound developed by Huang, Qian, Liu et al., called MGP, can drive the release of NO from cells.

To confirm the usefulness of their cells as a delivery system, Huang, Qian, Liu et al. transplanted some of the genetically modified mesenchymal stem cells into the kidneys of mice, and then showed that when these mice were given MGP, the levels of NO increased in the kidneys but not in other organs. This result confirms that the cell-based delivery system provides spatial and temporal control of the production of NO.

These findings demonstrate a new delivery system for therapies using gas molecules, which can be controlled spatiotemporally in mice. In the future, these types of systems could be used in the clinic for long-term treatment of conditions where artificial carriers could lead to complications.

## Introduction

Nitric oxide (NO), as a gaseous signaling molecule (GSM), plays vital roles in various physiological processes, including tissue regeneration (*Midgley et al., 2020*). Due to the extremely short half-life of NO, efforts to extensively enhance the therapeutic efficacy of NO by various types of artificial carriers, such as polymers, peptides, and nanoparticles, have achieved many successes, and there are remaining challenges that need to be addressed, including off-targeting, toxicity, immunogenicity, and clinical translation (*Wilhelm et al., 2016*; *Park, 2013*). Successful NO delivery requires an appropriate carrier for delivering these therapeutic molecules to the target sites in an on-demand and controlled manner, as well as in a protected, pharmacologically active form (*Weaver et al., 2014*). Bioinspired delivery vehicles, including biological cells, exosomes, and isolated membrane ghosts, are highly attractive because they exhibit excellent biodistribution, immune compatibility, innate disease-targeting abilities, and reduced toxicity (*Bush et al., 2021*; *Yoo et al., 2011*; *Samanta et al., 2018*; *Fang et al., 2018*). Among these systems, mesenchymal stem cells (MSCs) have gained considerable attention as drug-delivery vehicles owing to their genetic tractability, payload diversity, intrinsic tropism for disease sites, and differentiation potential (*Tran and Damaser, 2015*; *Thanuja et al., 2018*).

MSCs are adult stem cells capable of self-renewal and multilineage differentiation that have been reported to promote tissue regeneration mediated by immunomodulatory and proangiogenic properties via paracrine effects, and MSC-based cell therapeutics have entered multiple clinical trials (*Baraniak and McDevitt, 2010*). Previous studies revealed that MSCs do not express blood-group antigens or MHC class II antigens and have long been reported to be hypoimmunogenic or immune privileged, which possess the clinical potential of MSC-based cell therapy (*Zhou and Shi, 2023*). MSCs as delivery vehicles have been widely studied for the delivery of various drugs and bioactive molecules into target disease sites, including peptides, proteins, DNA, and RNA (*Peng et al., 2012*; *Zhang et al., 2021*). However, the concept of MSCs as vehicles to deliver GSMs has not been studied. In this regard, we propose an MSC-based gas-generating platform as a unique and tunable platform for extensively broad GSM therapies.

To achieve a gas-generating platform based on living cells, MSCs are programmed to sustain the generation and release NO by taking advantage of enzyme prodrug therapy, a versatile and exploitable technique to convert inactive, nontoxic prodrugs to active drugs at the desired sites (*Zhang et al., 2015*). In the present study, we developed an engineered MSC (eMSC)-based NO delivery platform for controlled NO delivery. In this advanced delivery platform, the expression of mutant β-galactosidase (β-GAL^H363A) by eMSCs enabled the production of NO when the NO prodrug was administered. We hypothesized that eMSCs could successfully generate and release NO in a precise spatiotemporal manner, as well as augment the therapeutic effects of MSCs and result in superior efficacy of stem cell therapy.

## Results

### MSC-mediated NO release platform

Other than mentioned in context, MSCs represent human placenta-derived MSCs. To establish a living cell-mediated gas generation platform, MSCs were genetically modified to produce mutant β-galactosidase (β-GAL^H363A), which triggers NO release from MSCs when the corresponding NO prodrug is applied and avoids the interference of endogenous glycosidase. We constructed a lentiviral vector to stably express mutant β-GAL^H363A, Renilla luciferase (Rluc), and red fluorescence protein (RFP), in which Rluc and RFP were used for molecular imaging and immunohistology, respectively (*Figure 1A*). The expression of β-GAL^H363A in eMSCs was confirmed (*Figure 1B and C* and *Figure 1—figure supplement 1A*). Moreover, Rluc expression was confirmed by in vitro molecular imaging (*Figure 1—figure supplement 1B*). Furthermore, compared with wild-type MSCs, eMSCs did not exhibit significant differences in morphology or MSC surface markers (*Figure 1—figure supplement 2*).

Based on our previous report (*Hou et al., 2019*), we designed a new NO prodrug, 6-methyl-galactose-benzyl-oxy NONOate, which was named MGP. MGP contains a lipid-soluble self-decomposition chain oil that enhances its water distribution coefficient and enables it to cross the cell membrane of eMSCs and then be catalyzed by β-GAL^H363A to release NO from eMSCs. The structure and synthesis routes of MGP are shown in *Figure 1D* and *Figure 1—source data 1*. To verify whether the redesigned NO prodrug MGP could release NO specifically by β-GAL^H363A catalysis, we measured the NO release profile of MGP in response to β-GAL^H363A in vitro. As shown in *Figure 1—figure supplement 3*, MGP can be specifically catalyzed by β-GAL^H363A and has sustained NO release behavior. In this novel NO release eMSC (NO-eMSC) system, eMSCs, as a manufacturing factory for producing β-GAL^H363A, could further release NO by catalyzing the NO prodrug MGP (*Figure 1E*). We detected NO production in the medium of eMSCs with MGP administration using the Griess method, confirming that MGP can diffuse into eMSCs and release NO (*Figure 1F*). We next directly measured the NO levels of eMSCs in vitro by an electron paramagnetic resonance (EPR) technique using the spin trap Fe•(DETC)2 colloid solution. The characteristic triplet EPR signal was observed in eMSCs with MGP administration (*Figure 1G*), and we further confirmed that $1 \times 10^6$ eMSCs could release 1 nmol NO (*Figure 1H*). Moreover, NO production of eMSCs with MGP administration was visualized by using diaminofluorescein (DAF)-FM diacetate, a cell-permeable fluorescent probe for the detection of intracellular NO, which also supported that eMSCs could specifically release NO (*Figure 1I and J*).

Considering the cell-protective effects of a low NO concentration, our results revealed that the optimum concentration of MGP was 2 μg/mL by cell viability, immunostaining for the proliferation marker Ki67, and immunostaining for the expression of proliferation-related genes (*Figure 1—figure supplement 4*). We further verified the protective effects of NO in response to oxidative stress-induced eMSC apoptosis. The bioluminescence imaging (BLI) assay showed that eMSC proliferation was markedly ameliorated by $H_2O_2$-induced oxidative stress (*Figure 1—figure supplement 5*).

### Enhanced antioxidation properties of eMSCs

To further investigate the transcriptomic profile of eMSCs with MGP administration, mRNA sequencing analysis was performed. RNA sequencing (RNA-seq) profiles from eMSCs and eMSCs with MGP administration passed quality control (*Figure 2—figure supplement 1*). We identified a total of 228 differentially expressed genes (DEGs), of which 116 were upregulated genes and 112 were downregulated genes (*Figure 2—figure supplement 2A*). Analysis of Gene Ontology (GO) categories was mainly enriched in biological processes, including protein transport, cell differentiation, morphogenesis, and

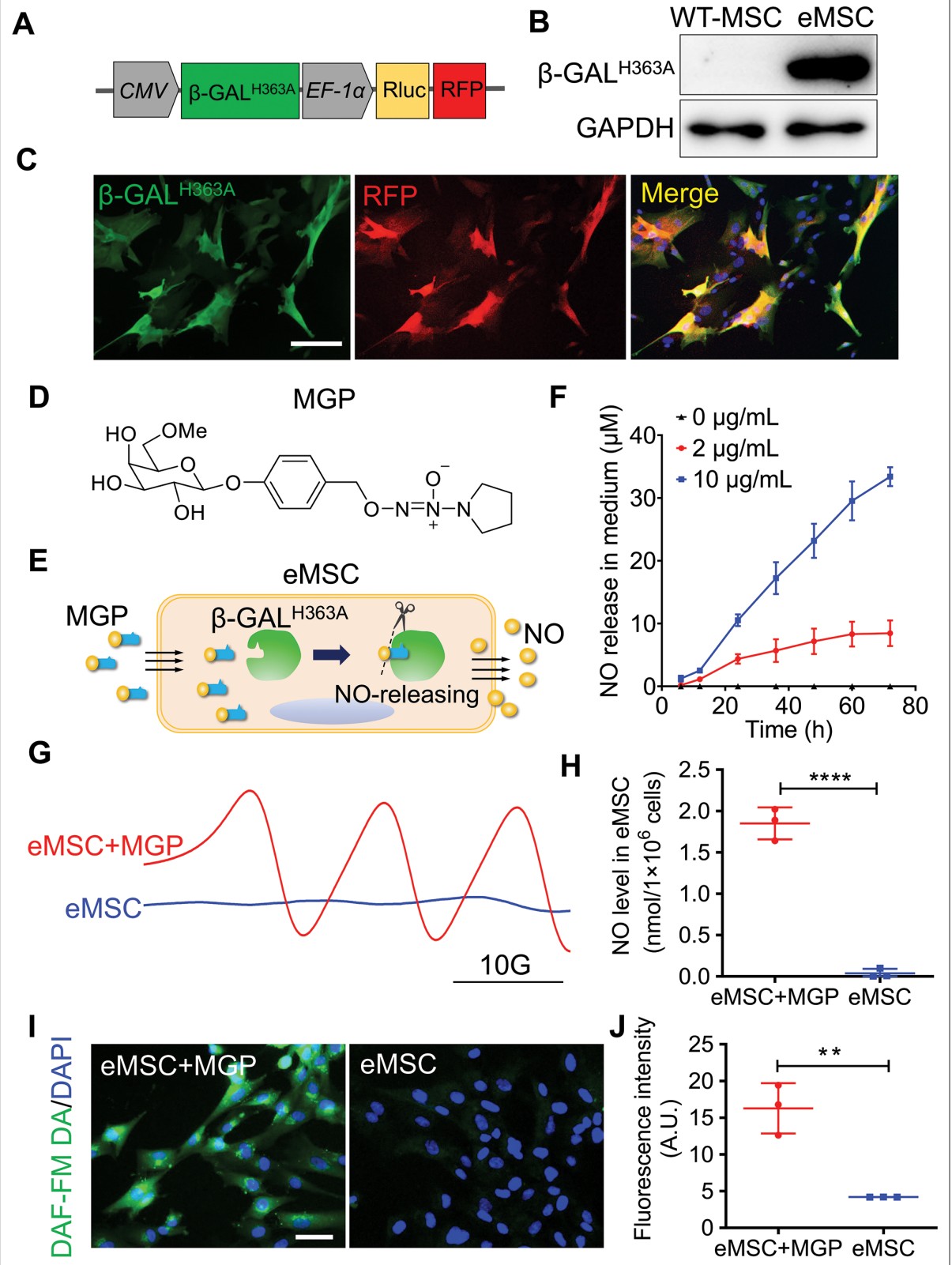

**Figure 1.** Evaluation of the nitric oxide (NO)-engineered mesenchymal stem cell (eMSC) system in NO generation and release. (**A**) Genetically modifying placenta-derived MSCs to express mutant β-galactosidase (β-GAL$^{H363A}$) through a lentiviral transduction system, thus enabling continuous production of β-GAL$^{H363A}$ for NO delivery. Moreover, the MSCs also stably expressed Renilla luciferase (Rluc) and red fluorescence protein (RFP) for in vivo bioluminescence imaging and immunofluorescence analysis, respectively. (**B**) The protein expression of β-GAL$^{H363A}$ in eMSCs was evaluated

*Figure 1 continued on next page*

*Figure 1 continued*

by 6×His tag protein detection due to β-GAL$^{H363A}$ carrying a C-terminal 6×His tag. (**C**) Immunofluorescence staining of β-GAL$^{H363A}$ (green) in eMSCs. Scale bars, 50 μm. (**D**) Structure of the NO prodrug MGP, 6-methyl-galactose-benzyl-oxy NONOate. (**E**) Construction of the NO-eMSC system and the mechanism of NO release from this system. (**F**) The NO release profile from eMSCs administered at different concentrations of NO-prodrug MGP was determined using the Griess assay. n=3. (**G**) NO release from eMSCs with MGP administration in vitro was assessed by electron paramagnetic resonance (EPR), and the characteristic triplet EPR signal was observed. eMSCs served as controls. (H) The amount of NO-Fe (DETC)$_2$ was calibrated using TEMPO as a standard. A total of $1 \times 10^6$ eMSCs approximately released 1 nmol NO in 1 hr. n=3. (**I**) The intracellular NO release in eMSCs treated with MGP was measured by diaminofluorescein (DAF) staining, in which DAF-FM DA is a cell-permeable fluorescent probe for the detection of intracellular NO. Scale bar, 25 μm. (**J**) Quantification of DAF fluorescence intensity in eMSCs in the presence of 5 μg/mL MGP for 6 hr. eMSC+MGP, eMSCs with MGP administration. All data are presented as the mean ± SD, **p<0.01, ***p<0.001, ****p<0.0001.

The online version of this article includes the following source data and figure supplement(s) for figure 1:

**Source data 1.** Synthesis route of the NO-prodrug MGP.

**Figure supplement 1.** Transduction of hP-mesenchymal stem cells (MSCs) with lentiviral vectors carrying β-GAL$^{H363A}$, Renilla luciferase (Rluc), and red fluorescence protein (RFP).

**Figure supplement 2.** Characterization of the engineered mesenchymal stem cell (eMSC).

**Figure supplement 3.** The redesigned NO-prodrug MGP could release NO specifically by β-GAL$^{H363A}$ catalysis.

**Figure supplement 4.** NO release facilitated the proliferation potential of engineered mesenchymal stem cells (eMSCs) in vitro.

**Figure supplement 5.** The survival of engineered mesenchymal stem cells (eMSCs) and eMSCs with MGP administration upon exposure to 200 μM H$_2$O$_2$ for 48 hr was analyzed by bioluminescence imaging (BLI).

oxidation-reduction processes (*Figure 2A*). Among them, the oxidation-reduction process attracted our attention, suggesting that NO may improve intracellular antioxidant capacity, which was confirmed by gene set enrichment analysis (GSEA) (*Figure 2B* and *Figure 2—figure supplement 2B*). Meanwhile, we noted that a significant increase in some of the DEGs associated with antioxidation was observed in eMSCs with MGP administration, namely, *GSR*, *SRXN1*, *RRBP1*, *BMI1*, *RPL24P4*, and *MAFG*, indicating that NO may indeed play a cytoprotective role in eMSCs by elevating antioxidation capacity (*Figure 2C*). We also found that NO led to robust increases in survival- or proliferation-related genes, including *FER*, *NRSN2*, *UCK2*, *CAB39*, *SNHG4*, and *SYDE1* (*Figure 2D*), as well as angiogenesis-related genes, such as *ARRB2*, *RHOQ*, *MYADM*, and *CFHR1* (*Figure 2—figure supplement 2C*). Consistent with the aforementioned findings, GSEA of the whole transcriptome also revealed negative regulation of cell apoptosis and positive regulation of angiogenesis by NO (*Figure 2E*, *Figure 2—figure supplement 2D*). In addition, Kyoto Encyclopedia of Genes and Genomes (KEGG) analysis highlighted pathways that may contribute to enhanced antioxidant capacity, including the Jak-STAT signaling pathway and MAPK signaling pathway (*Figure 2—figure supplement 2E*). Together, these results demonstrated that NO might be able to ameliorate the oxidative stress of eMSCs and inhibit cell apoptosis (*Figure 2F*).

## Enhanced renoprotection of eMSCs

To further validate whether NO could promote eMSC survival in ischemia/reperfusion (I/R)-injured kidneys, BLI analysis was performed for the real-time longitudinal monitoring of eMSC survival. A robust BLI signal of eMSCs was observed, indicating successful eMSC transplantation (*Figure 3A*). Although all groups gradually experienced donor cell death in the following days, eMSCs with MGP administration strikingly exhibited increased cell retention and prolonged cell survival in comparison with the eMSC group (*Figure 3B*). Furthermore, immunohistology also confirmed that the eMSCs with MGP administration displayed higher cell proliferation and cell retention on day 3 (*Figure 3C*). Moreover, we found that the eMSCs treated with MGP showed significantly lower levels of SCr and BUN (*Figure 3D and E*). Further kidney histology examination showed that tubular dilation, cast formation, and massive loss of brush borders at the initial stage after injury (3 days) were obviously reduced after NO-eMSC system treatment (*Figure 3F to H*). Kidney fibrosis analysis demonstrated that the development of interstitial fibrosis was consistent with the mRNA expression of ECM synthesis- and fibrosis pathway-related genes (*Figure 4*).

Furthermore, the expression of KIM-1, a kidney injury marker, was significantly decreased in the proximal tubules of the kidneys in the eMSC with MGP administration group compared to the eMSC group (*Figure 3—figure supplement 1A and B*). Simultaneously, we found that the level of caspase-3/

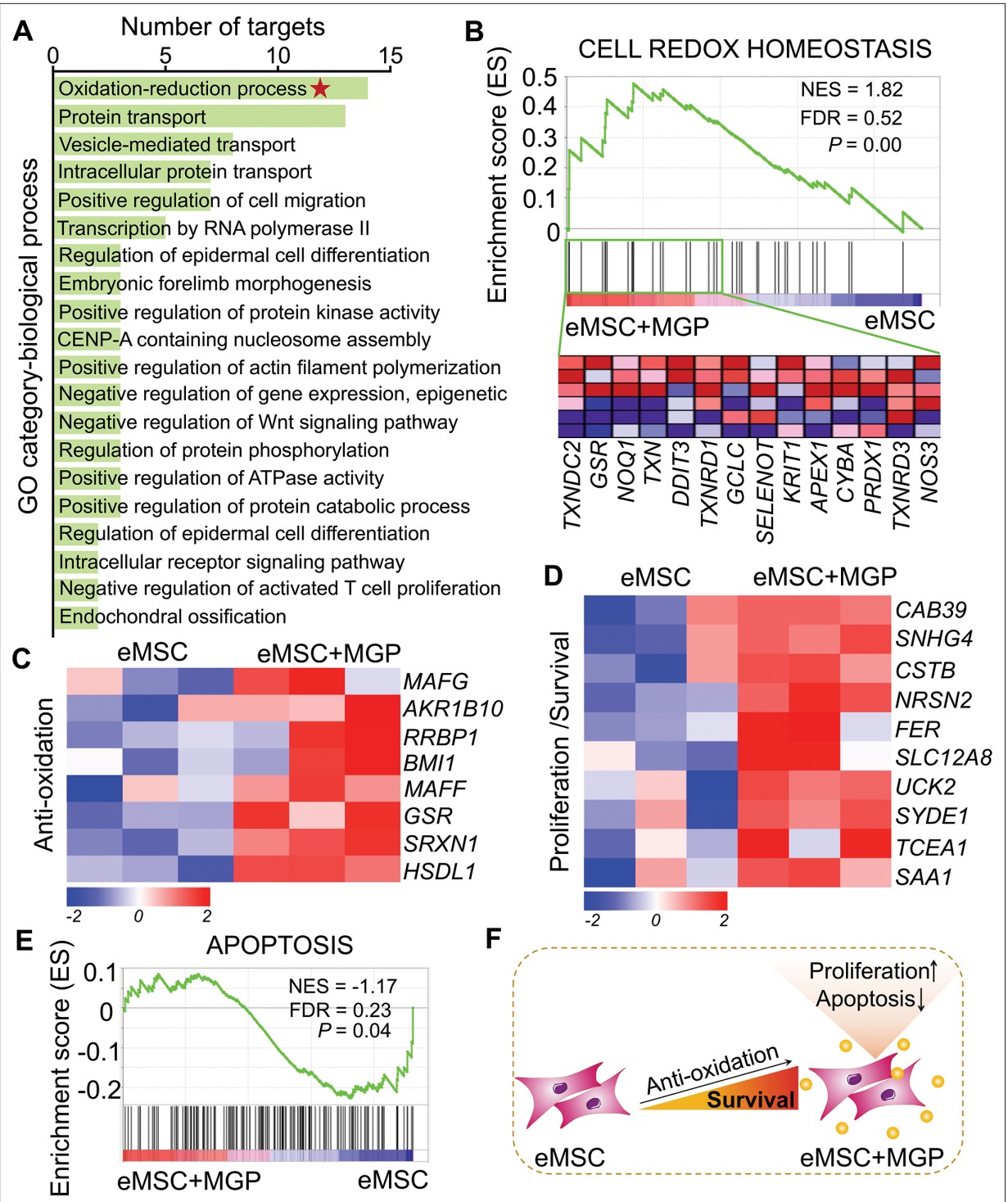

**Figure 2.** RNA sequencing (RNA-seq) analysis revealed that nitric oxide (NO) improves the antioxidant capacity of engineered mesenchymal stem cells (eMSCs). (**A**) Gene Ontology (GO) category analysis of differentially expressed genes (DEGs) for biological processes. Only significantly enriched terms with corrected p<0.05 are indicated. The top 20 enriched biological processes are ranked by the number of DEGs. (**B**) Gene set expression analysis (GSEA) revealed enrichment for the cell redox homeostasis pathway. The normalized enrichment score (NES), false discovery rate (FDR), and p value are indicated in the insert. (**C, D**) Heatmap of representative antioxidation (**C**) and proliferation/survival (**D**)-related genes. Fold change >1.5, p<0.05. Red indicates upregulation, and blue indicates downregulation. (**E**) GSEA revealed enrichment for apoptosis pathways. (**F**) Schematic representation of the prosurvival potential of the NO-eMSC system. eMSC+MGP, eMSCs with MGP administration.

The online version of this article includes the following figure supplement(s) for figure 2:

*Figure 2 continued on next page*

*Figure 2 continued*

**Figure supplement 1.** Quality control of RNA sequencing (RNA-seq).

**Figure supplement 2.** RNA sequencing (RNA-seq) analysis revealed that the nitric oxide (NO)-engineered mesenchymal stem cell (eMSC) system improved eMSC functionality.

cleaved caspase-3 exhibited an abrupt increase in the injured kidney of the PBS group but was attenuated by eMSCs with MGP administration (*Figure 3—figure supplement 1C and D*), suggesting that the NO-eMSC system has protective effects on apoptosis in tubular epithelial cells. Overall, these results indicated that the NO-eMSC system could increase eMSC survival in vivo and further improve kidney function.

## Evaluation of NO levels in the kidney

To further evaluate in vivo NO release from the eMSCs, eMSCs were transplanted into the left kidney by renal parenchymal injection followed by intravenous injection of the NO prodrug MGP. The results from Griess (*Figure 5A*) and chemiluminescence assays (*Figure 5B and C*) showed that the NO level from eMSC-treated kidneys was significantly increased when the prodrug MGP was applied. We next directly measured the NO levels of eMSC-treated kidneys in vivo by an EPR technique using the spin trap ferrous $N$-methyl-d-glucamine dithiocarbamate complex (($MGD)_2Fe^{2+}$). The EPR signal of NO was observed in all groups, while an obvious characteristic triplet EPR signal (NO signal) was observed in the eMSCs treated with MGP (*Figure 5D and E*), which further confirmed that eMSCs could release NO in vivo. Moreover, targeted NO delivery was evaluated by comparison of NO levels in different tissues of mice after the administration of the eMSC in the kidney. The quantitative data showed that NO production from the kidney was significantly higher than that from the heart and liver (*Figure 5F to H*), suggesting that eMSCs could release NO specifically in the kidney.

## Extended applications of this cell-based NO delivery platform

To investigate the universality of our NO delivery approach for applications in AKI, we conducted additional experiments to confirm whether other MSCs will perform equally well. Human adipose-derived MSCs (AD-MSCs) and human umbilical cord MSCs (hUC-MSCs) were genetically modified to produce mutant β-galactosidase (β-GAL$^{H363A}$). NO production by engineered AD-MSCs (eAD-MSCs) and engineered hUC-MSCs (ehUC-MSCs) with MGP administration in vitro was detected by the EPR technique. The characteristic triplet EPR signal was observed in eAD-MSCs and ehUC-MSCs with MGP administration (*Figure 5—figure supplement 1A and B*, *Figure 5—figure supplement 2A and B*), which was further confirmed by the Griess method (*Figure 5—figure supplement 1C and D*, *Figure 5—figure supplement 2C and D*). Next, the therapeutic effects of eAD-MSCs or ehUC-MSCs with MGP administration in AKI mice were estimated via histological analysis and renal function analysis. We found that the eAD-MSCs and ehUC-MSCs treated with MGP showed significantly lower levels of SCr and BUN (*Figure 5—figure supplements 1E and 2E*). Further kidney histology examination showed tubular dilation, cast formation, and massive loss of brush borders at the initial stage after injury (3 days) (*Figure 5—figure supplement 3A*). Furthermore, the expression of KIM-1 (kidney injury marker) was significantly decreased in the proximal tubules of the kidneys in the eAD-MSC and ehUC-MSC with MGP administration groups (*Figure 5—figure supplement 3B*). Thus, our MSC and NO prodrug system provides a tunable platform for NO delivery and holds promise for regenerative therapy.

## Enhanced proangiogenic effects of eMSCs

VEGFR2-Fluc transgenic mice were used to monitor renal angiogenesis in real time, which appeared as BLI signals. The schematic diagram is shown in *Figure 6—figure supplement 1*. According to the results of angiogenesis imaging, BLI signals were emitted in all groups, and the strongest signal was detected in the eMSCs with MGP administration group, which suggests that the NO-eMSC system could stimulate renal angiogenesis by activating the VEGF/VEGFR2 pathway (*Figure 6A–C*). We next verified neovascularization in damaged kidneys on day 7 post-surgery by histological examination. The CD31 immunostaining results revealed significantly enhanced CD31$^+$ microvascular density in the group of eMSCs with MGP administration, which is consistent with the BLI results (*Figure 6D*).

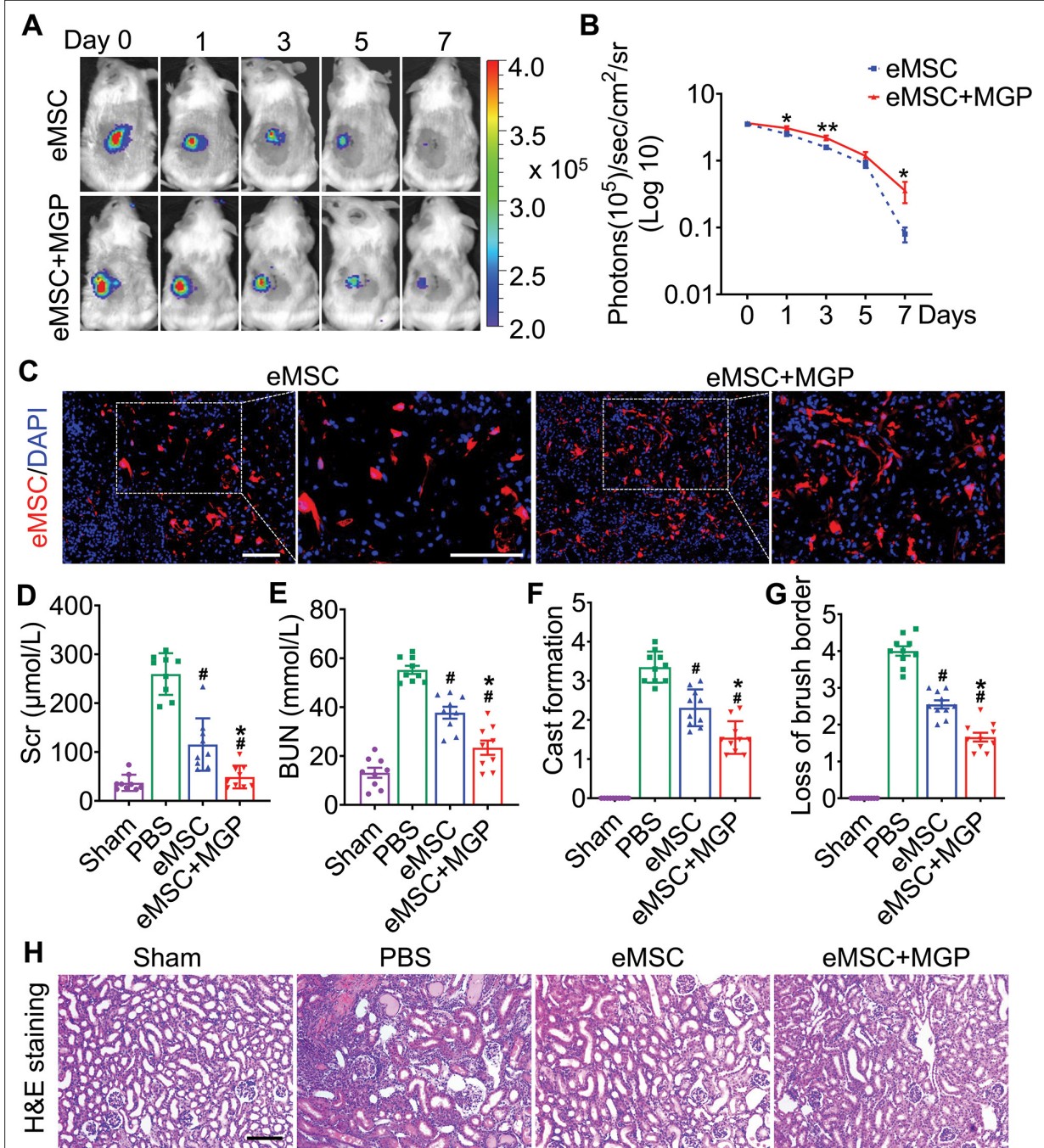

**Figure 3.** The nitric oxide (NO)-engineered mesenchymal stem cell (eMSC) system confers renoprotection by increasing eMSC survival in vivo. (**A**) The survival ratio of eMSCs and eMSCs with MGP administration after transplantation in ischemia/reperfusion (I/R)-induced acute kidney injury (AKI) model mice was tracked by bioluminescence imaging (BLI). Images are from representative animals receiving $1×10^6$ eMSCs alone or eMSCs with MGPs. (**B**) Quantitative analysis of BLI signals demonstrates that eMSCs with MGP administration could prolong cell survival. Data are presented as the mean ± SD, *$p<0.05$, **$p<0.01$. (**C**) Representative photomicrographs display the engraftment of eMSCs and eMSCs with MGP administration (RFP, red) within kidneys on day 3. Cell nuclei were stained with DAPI (blue). Scale bar, 50 µm. (**D and E**) Renal function indexes, including serum creatine (**D**) and blood urea nitrogen (**E**) levels, were measured on day 3 after AKI. n=9. (**F and G**) Semi-quantitative histological assessment of cast formation (**F**) and loss of brush border (**G**) of H&E staining on day 3 after AKI. n=10. (**H**) Representative images of H&E staining on day 3 post AKI. eMSC+MGP, eMSCs with MGP administration. Scale bar, 100 µm. All data are presented as the mean ± SD, eMSC+MGP, eMSCs with MGP administration. *$p<0.05$ versus eMSC; #$p<0.05$ versus PBS.

The online version of this article includes the following figure supplement(s) for figure 3:

*Figure 3 continued on next page*

*Figure 3 continued*

**Figure supplement 1.** The nitric oxide (NO)-engineered mesenchymal stem cell (eMSC) system protected against kidney injury by rescuing renal tubules.

Moreover, the expression of the angiogenesis-related genes *VEGFR2, bFGF, PLGF, Ang-1,* and *Ang-2* detected by real-time polymerase chain reaction (RT-PCR) also confirmed that the eMSCs with MGP administration facilitated renal angiogenesis by upregulating angiogenic factor expression (***Figure 6— figure supplement 2***). Recovery from renal I/R injury requires tubular cell proliferation to promote kidney regeneration. As shown in ***Figure 6E and F***, the numbers of proliferating Ki-67+ tubular cells were largely stimulated in the group of eMSCs treated with MGP, suggesting that the NO-eMSC system promoted kidney regeneration by accelerating tubular cell proliferation. The promotion of renal angiogenesis contributes to the restoration and repair of damaged kidneys, which paves the way for successful kidney regeneration.

Subsequently, we explored the mechanism of eMSCs with MGP administration-induced angiogenesis, and our data revealed that eMSCs boosted the migration ability of human umbilical vein endothelial cells (HUVECs) in a coculture system with Transwells, as evidenced by a scratch wound-healing assay (***Figure 6—figure supplement 3A and B***). Additionally, the angiogenic ability of HUVECs, as manifested by the number of nodes and branches, was visibly increased in the group of eMSCs treated with MGP, as assessed by a tube formation assay (***Figure 6—figure supplement 3C, D, and E***). Additionally, the protein level of cleaved caspase-3, a key apoptosis molecule, was detected

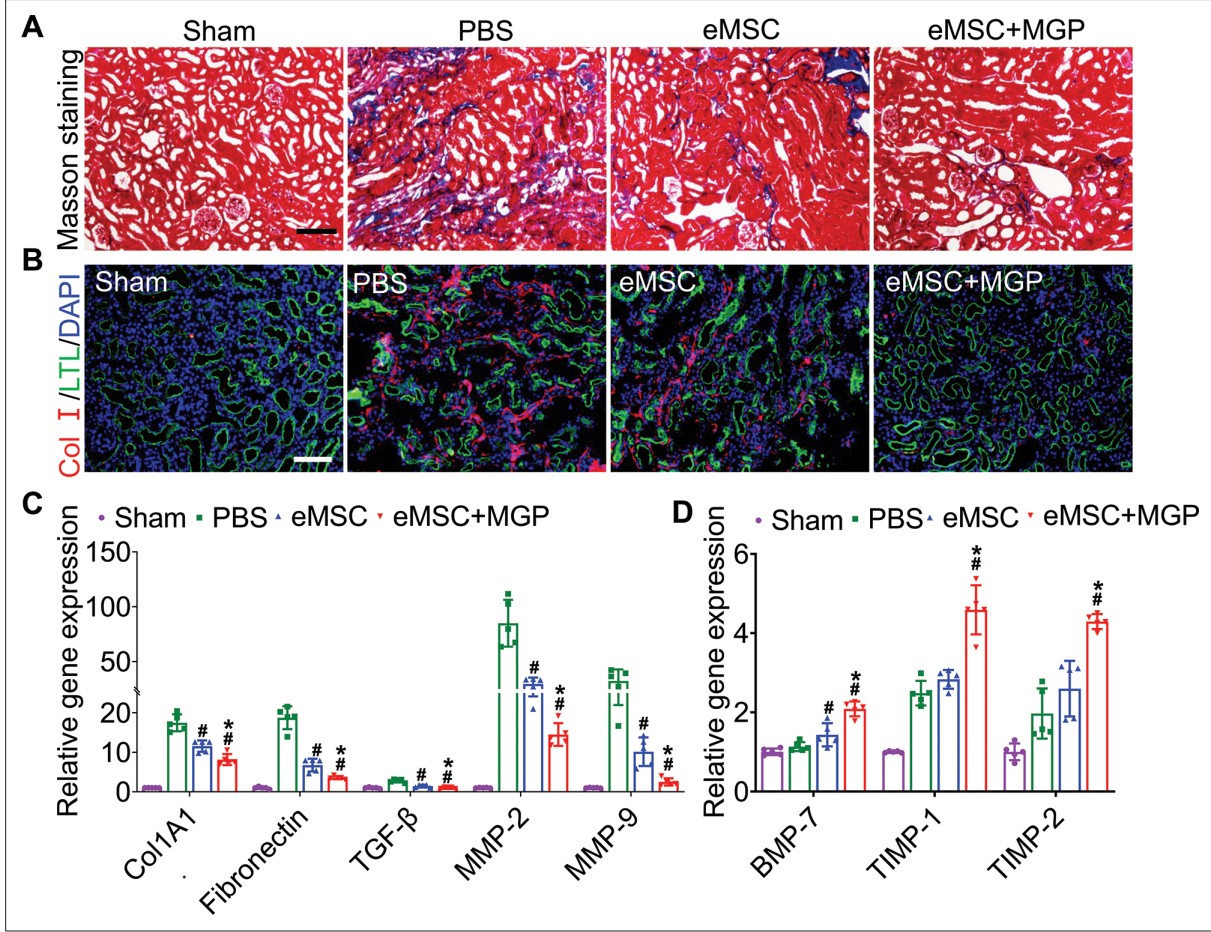

**Figure 4.** Engineered mesenchymal stem cells (eMSCs) with MGP administration attenuate renal fibrosis. (**A**) Masson trichrome staining was performed to examine kidney fibrosis on day 28 post acute kidney injury (AKI). Scale bar, 100 μm. (**B**) COL-I immunofluorescence staining was performed to examine kidney fibrosis on day 28 post AKI. Scale bar, 100 μm. (**C and D**) Real-time polymerase chain reaction (PCR) analysis of fibrosis-related gene expression in the kidney on day 28 after AKI. n=5. All data are presented as the mean ± SD, *p<0.05 versus eMSC; #p<0.05 versus PBS.

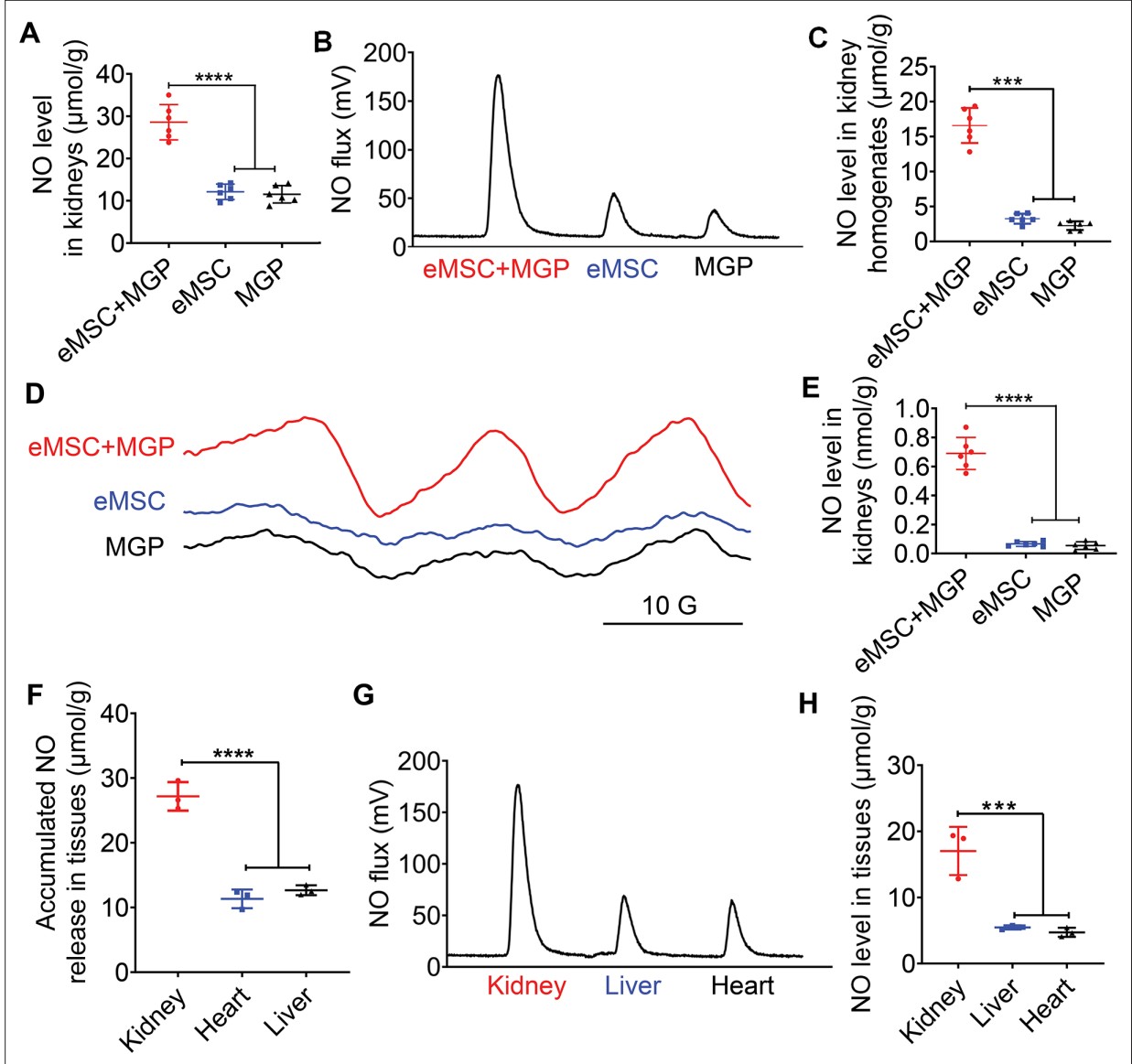

**Figure 5.** Detection of NO release from the NO-engineered mesenchymal stem cell (eMSC) system in vivo. (**A**) The Griess kit assay was used to measure NO levels in kidneys with MGP alone, eMSC alone, or eMSC treated with MGP. n=6. (**B**) Detection and quantification of NO levels in kidneys in vivo using chemiluminescence. The area of each peak represents the corresponding amount of NO. (**C**) Calculation of the NO concentration in each group from the standard curve. n=6. (**D**) NO release from eMSCs with MGP administration in kidneys was assessed by electron paramagnetic resonance (EPR). MGP and eMSC served as controls. (**E**) Quantitative analysis of NO production in each group. The amount of NO-Fe(DETC)$_2$ was calibrated using TEMPO as a standard. n=6. (**F**) A Griess kit assay was used to measure NO levels in various tissues of renal eMSC-transplanted mice. n=3. (**G**) Detection and quantification of NO levels in different tissues from renal eMSC-transplanted mice in vivo using chemiluminescence. The area of each peak represents the corresponding amount of NO. (**H**) Calculation of the NO concentration in each group from the standard curve. n=3. eMSC+MGP, eMSCs with MGP administration. All data are presented as the mean ± SD, ***p<0.001, ****p<0.0001.

The online version of this article includes the following figure supplement(s) for figure 5:

**Figure supplement 1.** The nitric oxide (NO)-engineered adipose-derived mesenchymal stem cell (eAD-MSC) system confers renoprotection.

**Figure supplement 2.** The nitric oxide (NO)-engineered human umbilical cord mesenchymal stem cell (ehUC-MSC) system confers renoprotection.

**Figure supplement 3.** Renoprotection of engineered adipose-derived mesenchymal stem cell (eAD-MSCs) and engineered human umbilical cord mesenchymal stem cells (ehUC-MSCs).

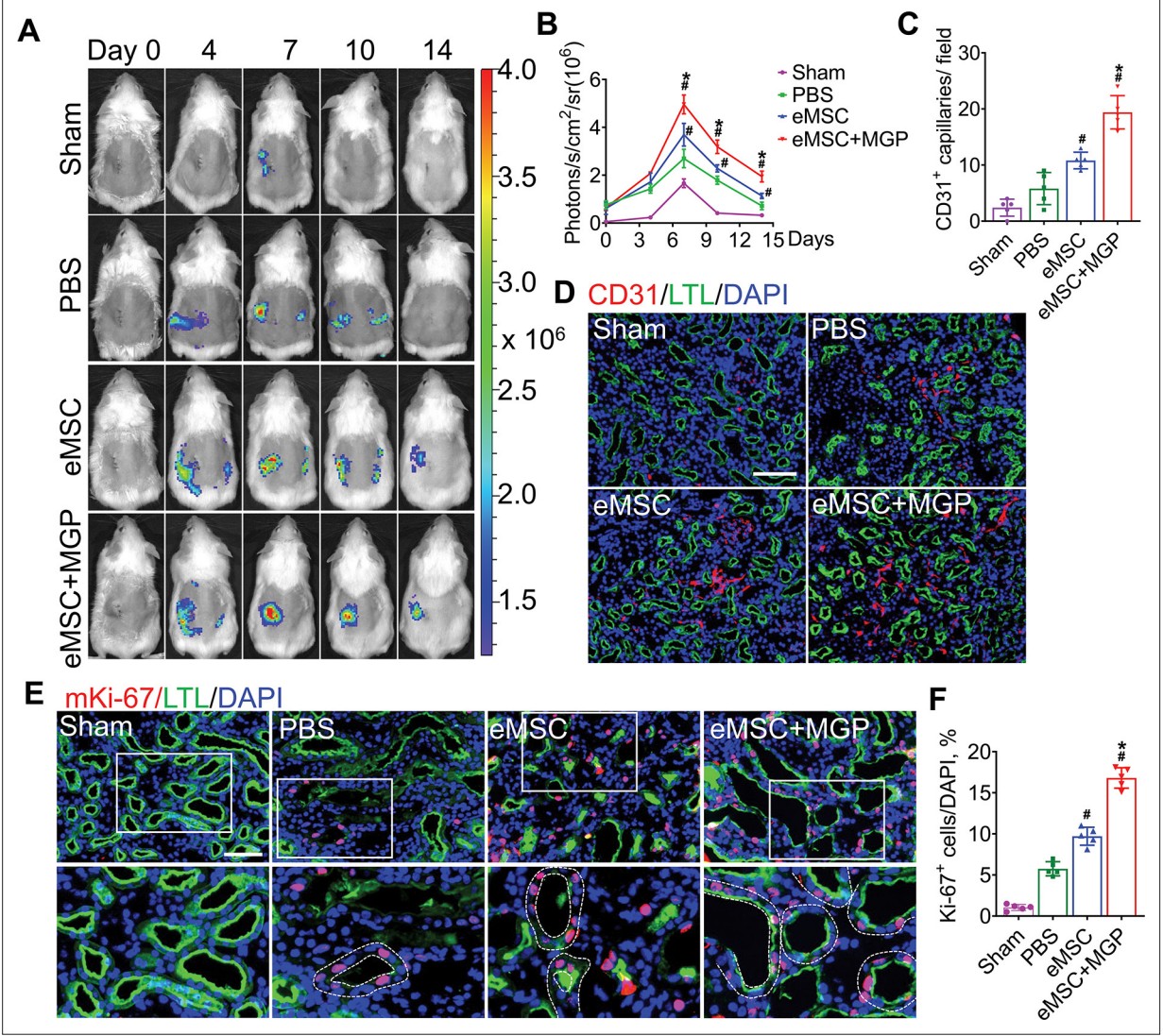

**Figure 6.** The nitric oxide (NO)-engineered mesenchymal stem cell (eMSC) system promoted kidney regeneration after acute kidney injury (AKI). (**A**) In vivo bioluminescence imaging (BLI) was used to monitor the spatiotemporal dynamics of VEGFR2 expression following eMSC administration in a mouse AKI model using Vegfr2-Fluc transgenic mice. (**B**) Quantification of BLI signals revealed that eMSCs with MGP could enhance angiogenesis in injured kidneys. The average radiance of Fluc was expressed as photons/s/cm²/sr. n=3. (**C**) Quantification of CD31-positive capillaries in injured kidneys on day 7 after AKI. CD31 is a marker of endothelial cells and can be used for angiogenesis evaluation. n=5. (**D**) Immunofluorescence staining of CD31 was performed on day 7 after AKI. Scale bar, 100 μm. LTL (green) was used to reveal proximal tubules. (**E**) Representative images of immunofluorescence staining of Ki67. (**F**) Quantification of the percentage of Ki67-positive cells in kidneys on day 3 after AKI. n=5. Scale bar, 50 μm. eMSC+MGP, eMSCs with MGP administration. All data are presented as the mean ± SD, *p<0.05 versus eMSCs; #p<0.05 versus PBS.

The online version of this article includes the following figure supplement(s) for figure 6:

**Figure supplement 1.** Schematic representation of VEGFR2-Fluc-KI transgenic mice to characterize angiogenesis.

**Figure supplement 2.** Enhanced angiogenic effects of the nitric oxide (NO)-engineered mesenchymal stem cell (eMSC) system.

**Figure supplement 3.** Engineered mesenchymal stem cells (eMSCs) with MGP administration enhanced the proangiogenic potential of human umbilical vein endothelial cells (HUVECs) in vitro.

by immunofluorescence staining, suggesting that eMSCs could inhibit the apoptosis of HUVECs (*Figure 6—figure supplement 3F*). In general, we proposed that eMSCs with MGP administration exert superior promotive effects on kidney angiogenesis after injury by enhancing the proangiogenic activities of vascular endothelial cells.

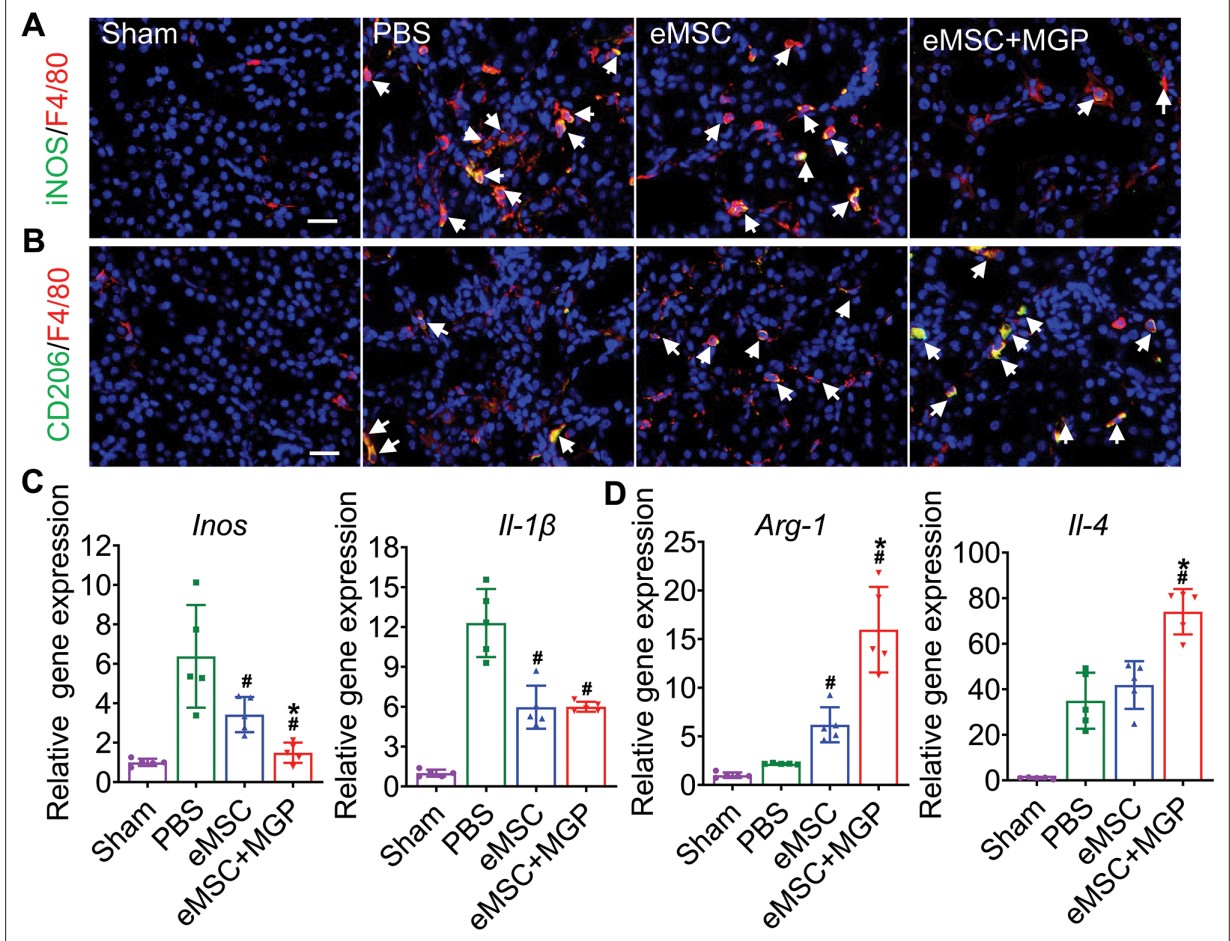

**Figure 7.** The nitric oxide (NO)-engineered mesenchymal stem cell (eMSC) system modulates the inflammatory response by reshaping macrophage polarization. (**A**) Immunofluorescence images by costaining for the M1 macrophage marker iNOS (green) and pan-macrophage marker F4/80 (red) in injured kidneys on day 3 after acute kidney injury (AKI). M1 macrophages are indicated by white arrows. Scale bar, 50 μm. (**B**) Immunofluorescence images by costaining for the M2 macrophage marker CD206 (green) and pan-macrophage marker F4/80 (red) in injured kidneys on day 3 after AKI. M2 macrophages are indicated by white arrows. Scale bar, 50 μm. (**C**) Real-time polymerase chain reaction (RT-PCR) analysis of M1 macrophage-related gene (iNOS and IL-1β) expression in kidney tissues on day 3 after AKI. n=5. (**D**) RT-PCR analysis of M2 macrophage-related gene (IL-4 and Arg-1) expression in kidney tissues on day 3 after AKI. n=5. eMSC+MGP, eMSCs with MGP administration. All data are presented as the mean ± SD, *p<0.05 versus eMSCs; #p<0.05 versus PBS.

The online version of this article includes the following figure supplement(s) for figure 7:

**Figure supplement 1.** The nitric oxide (NO)-engineered mesenchymal stem cell (eMSC) system ameliorates oxidative stress by protecting against antioxidation.

## Diminished modulated inflammatory responses

We next explored the modulatory effect of eMSCs on the inflammatory response in I/R-injured kidneys, and our results showed that eMSCs with MGP administration increased the number of F4/80- and CD206-positive macrophages while decreasing the number of F4/80- and iNOS-positive macrophages, indicating that the NO-eMSC system could promote the polarization of M1 macrophages to M2 macrophages (**Figure 7A and B**). Meanwhile, mRNA expression profiling further confirmed this finding (**Figure 7C and D**). Oxidative stress-induced reactive oxygen species (ROS) are well known as the main cause of renal I/R injury, leading to subsequent augmented inflammation and extended tissue damage (**Collard and Gelman, 2001**). Damaged renal tubular cells were overloaded with excessive lipids caused by abnormal lipid metabolism, while elevated ROS levels can lead to irreversible oxidative damage to lipids and further result in cell damage and death. Excessive intracellular accumulation of lipids was observed in the renal tubules of the PBS group after I/R injury, while the eMSCs with MGP administration significantly decreased lipid deposition (**Figure 7—figure supplement 1A**

*and B*). Meanwhile, the renal level of MDA, an index of ROS-mediated lipid peroxidation, was notably increased in the PBS group and significantly decreased in the eMSCs with MGP administration group (*Figure 7—figure supplement 1C*). Moreover, we examined the antioxidant capacity levels (SOD and GSH contents) in kidney tissues, and remarkable upregulation of SOD and GSH levels was observed in the group of eMSCs treated with MGP (*Figure 7—figure supplement 1D and E*). In conclusion, the NO-eMSC system exhibited a renoprotective effect by targeting antioxidation and macrophage polarization-mediated anti-inflammation.

## Discussion

In this study, we developed an eMSC-mediated NO gas-generating platform, the NO-eMSC system, for site-specific, controlled, and long-term delivery of NO for the first time. This advanced NO-eMSC system was engineered to both program the continuous release of NO and enhance stem cell-mediated regeneration of damaged tissues, which is a unique and mosaic approach to maximize the therapeutic effect. The NO-eMSC system successfully generated and released NO in a precise spatio-temporal manner and improved eMSC functionalities, resulting in higher cell survival and more resistance to apoptosis under oxidative stress. The NO-eMSC system exerted superior therapeutic effects, including rescuing renal tubules, relieving inflammation, stimulating angiogenesis, and improving renal function.

GSMs with pharmaceutical effects, such as oxygen, NO, carbon monoxide, hydrogen, and hydrogen sulfide ($H_2S$), have tremendous potential in the treatment of many diseases (*Ji et al., 2016*; *Hu et al., 2021*; *Markhard et al., 2022*). Notably, the therapeutic effects of GSMs are highly concentration dependent; for example, low levels of NO (nmol to μmol) are frequently associated with cell protection, while high concentrations (>mmol) of NO exert apoptotic effects (*Midgley et al., 2020*). Although a plethora of GSM small molecule donors have been developed, the insufficient solubility, poor stability, and unfavorable pharmacokinetics of these GSM donors have prevented their wide application (*Hu et al., 2021*). NO, the first and most well-known gaseous molecule, has been widely shown to play critical roles in a wide spectrum of physiological and pathological processes (*Hu et al., 2021*; *Wang et al., 2015*). As a result, many kinds of NO-releasing platforms have emerged in the field of chemical materials to deliver NO in a controlled manner (*Hu et al., 2021*; *Zafonte et al., 2022*). Although endogenous (enzyme and pH variations) or exogenous (light and ultrasound)-triggered NO delivery systems for controlled NO release profiles have been developed, it remains a challenge to maintain long-term NO release and eliminate immunogenicity and systemic toxicity (*Zafonte et al., 2022*; *Shimizu et al., 2015*).

In recent years, an increasing number of cell delivery systems have entered multiple clinical trials or been approved for marketing (*Jackson et al., 2016*; *Hartmann et al., 2017*). Genetic engineering of MSCs is an approach to produce and deliver specific therapeutic proteins or enhance innate properties such as migration, differentiation, and survival for treating many tremendous diseases (*Park et al., 2015*). Therefore, genetically eMSCs, usually as living 'drug reservoirs', offer long-term and stable production of the therapeutic protein of interest and thus have gained appreciable attention as therapeutic delivery vehicles (*Bush et al., 2021*). Taking advantage of this NO delivery system and stem cell therapy, we engineered MSCs as an enzyme reservoir that catalyzes the NO prodrug to produce NO on demand, which combines the therapeutic potential of MSCs and NO.

For NO delivery, the undesired release of NO was observed due to the widespread distribution of endogenous β-GAL in blood and other tissues (*Hou et al., 2019*). Our previous study developed a novel NO-enzyme-prodrug pair for the specific release of NO, in which native β-GAL is transformed into mutant homologs with a 'hole' and only interacts with its corresponding 'bump'-prodrug (*Hou et al., 2019*). Hence, genetically eMSCs as living reservoirs enabled the continuous production of the 'hole'-enzyme (β-GAL$^{H363A}$), thereby achieving the controlled release of NO from its corresponding NO-prodrug, MGP. Moreover, the NO detection results confirmed that our NO-generating platform could release NO in a precise spatiotemporal manner; the duration and dosing of NO release could be tuned by controlling the NO prodrug. Specifically, NO release is initiated in the presence of the NO prodrug as long as eMSCs survive. NO dosing is controlled by the concentration of the administered NO prodrug, and an elevated NO signal was observed only at the site of transplanted eMSCs. In this study, we substantially advance the concept of a cell-mediated gas-generating platform for NO long-term applications, which may be applied for other gas enzyme prodrug pairs in the future.

The combinational use of NO-releasing biomaterials and stem cells is an emerging field of regenerative medicine (*Wang et al., 2015*); however, temporal and local release of NO-releasing biomaterials results in less contact time with cells and hinders their long-term applications. Moreover, in-depth knowledge of the underlying mechanisms to elucidate the influence of NO on stem cells and the stem cell microenvironment is of great value in the fields of biomaterials and regenerative medicine (*Midgley et al., 2020*; *Wang et al., 2015*). In this study, we developed an advanced MSC-mediated NO-generating platform engineered to both program the continuous release of NO and enhance stem cell-mediated regeneration of damaged tissues. Moreover, the NO-eMSC system exerted an immuno-suppressive effect by reshaping macrophage inflammatory properties and an enhanced angiogenesis effect to promote kidney regeneration. This endogenous-like NO-generating platform may represent a new paradigm for the future of long-term NO application on MSCs or other cell types.

In conclusion, we have developed a genetic eMSC-based NO delivery, NO-eMSC system, for targeted, controlled, and long-term NO delivery for the first time. This advanced NO delivery system can serve as a generic delivery system that holds promise for extensively broadening other gaseous molecule therapies and provides a new library of clinical solutions for regenerative medicine.

# Materials and methods
## Cell culture
### HUVEC culture
Primary HUVECs were purchased from the American Type Culture Collection (ATCC, Manassas, VA, USA) and grown in EGM2 medium (Lonza, Walkersville, MD, USA). Cells from passages 4–7 were used for the experiments in this study.

### HK-2 culture
A human proximal tubule epithelial cell line (HK-2) was purchased from ATCC and cultured in DMEM/F12 medium (Gibco) supplemented with 10% FBS, penicillin (100 U/mL), and streptomycin (100 µg/mL).

### HEK-293T culture
Human embryonic kidney 293T cells (HEK 293T) were obtained from ATCC and maintained in high-glucose DMEM with 10% FBS penicillin (100 U/mL) and streptomycin (100 µg/mL).

### MSC culture
Human placenta-derived MSCs, human AD-MSCs, and hUC-MSCs were purchased from AmCell-Gene (Tianjin, China). All MSCs were cultured in Dulbecco's modified Eagle's medium/F-12 (DMEM/F12) nutrient mixture (Gibco) supplemented with fetal bovine serum (10%, Gibco) and 100 units/mL penicillin-streptomycin (100×, Gibco) as previously described (*Liang et al., 2017*). The MSCs were transduced with lentiviral vectors carrying β-GAL$^{H363A}$ (*Hou et al., 2019*), Rluc, and RFP. Stable cells were isolated using FACS for RFP expression. The Rluc activity of eMSCs was confirmed in vivo and in vitro using the IVIS Lumina II system (Xenogen Corporation, Hopkinton, MA, USA). The cell surface marker expression of eMSCs was tested using a FACSCalibur flow cytometer (BD Biosciences).

## Construction of plasmids and eMSCs
The coding sequence of mutant β-galactosidase (β-GAL$^{H363A}$) can be obtained from a previous report (*Hou et al., 2019*). To achieve localization of the transplanted MSCs, we cloned the coding sequence of double-fusion reporter genes (Rluc-RFP, Rluc, and red fluorescent protein) from a previously reported pcDNA3.1-RFP-Rluc-HSV-ttk plasmid (*Cao et al., 2007*; *Leng et al., 2014*). Subsequently, the sequences encoding β-GAL$^{H363A}$ and Rluc-RFP were inserted into the lentiviral backbone plasmid, which was carried out by Miaolingbio Company (Wuhan, China). The lentiviral vector was cotransfected with the second-generation lentivirus packing vectors psPAX2 and pVSVG into 293T cells by Lipofectamine 2000 (Invitrogen, Grand Island, NY, USA) to produce lentivirus. The infection efficiency was evaluated by RFP expression, as directly observed under an inverted fluorescence microscope

(Olympus, Japan), and further confirmed by β-GAL[H363A] expression (C-His-tagged recombinant protein), as evidenced by immunofluorescence.

## Synthesis route of the NO prodrug MGP

All chemicals and reagents were purchased from Sigma-Aldrich (China-mainland) and Energy Chemical (China-mainland) and used directly without further purification. All solvents were distilled before use, including DCM (CaH$_2$) and acetone (4 Å MS). Dry DMF, CH$_3$CN, and CH$_3$OH were purchased from Innochem (Beijing, China). $^1$H NMR and $^{13}$C NMR spectra were recorded on a Bruker Avance-400 FT nuclear magnetic resonance spectrometer. Chemical shifts were reported relative to the reference chemical shift of the NMR solvent. The synthesis route of MGP from compound 1 to compound 6 (MGP) is described in *Figure 1—source data 1*.

## Western blotting analysis

MSCs were collected and homogenized in RIPA lysis buffer containing protease inhibitors (Solarbio, Shanghai, China), and a BCA Protein Assay Kit (Thermo Scientific) was used to quantify the total protein. The total proteins were diluted in 4× SDS-PAGE loading buffer, boiled for 10 min, electrophoresed on 10% polyacrylamide gels, and blotted on 0.2 μm polyvinylidene fluoride membranes (Millipore, Darmstadt, Germany). After blocking with 5% nonfat milk for 2 hr at room temperature, the blots were incubated with primary antibodies overnight at 4°C followed by secondary antibodies at room temperature for 2 hr. An anti-6×His tag (β-GAL[H363A]) antibody was used. Signals were generated by using enhanced chemiluminescence reagent (Millipore, USA) and were captured by using a Tanon-5200 Chemiluminescence Imaging System (Tanon Science & Technology Co Ltd., Shanghai, China). Quantification was performed using ImageJ software. Antibodies are listed in *Supplementary file 1*.

## Flow cytometry analysis

For eMSC apoptosis analysis, 1×10$^4$ eMSCs were cultured on 12-well plates and then treated with 200 μM hydrogen peroxide (H$_2$O$_2$) for 12 hr. Cells were collected and then costained with annexin V-fluorescein isothiocyanate (Annexin V)-FITC (CA1020, Solarbio) and 7-aminoactinomycin D (7-AAD) (ST515, Beyotime) for 30 min, and cell apoptosis rates were measured using a FACSCalibur flow cytometer (BD Biosciences) and analyzed using FlowJo software (Tree Star).

## Bulk RNA-seq

RNA-seq was performed by LC Bio Tech (Hanzghou, China). Briefly, eMSCs treated with MGP (2 μg/mL) or PBS cultured for 24 hr were collected. Total RNA of eMSCs and eMSCs with MGP administration was extracted using TRIzol reagent (Invitrogen). The integrity of the RNA was demonstrated with a Bioanalyzer 2100 (Agilent, CA, USA), RIN >7.0, and confirmed by denaturing agarose gel electrophoresis. Total RNA was sent for RNA-seq library construction and high-throughput sequencing on an Illumina Novaseq 6000 with 2×150 bp paired-end sequencing (PE150). After removing low-quality and undetermined bases using Cutadapt software, we mapped the reads to the genome using HISAT2 software. DEGs among the different samples were defined by the DESeq2 package as criteria of fold change (FC)≥1.5 or FC ≤0.67 and p-value ≤0.05. Functional enrichment analyses of GO and KEGG were based on the Gene Ontology Database (http://www.geneontology.org/) and KEGG pathway database (http://www.genome.jp/kegg/), respectively. GSEA/MSigDB databases were used for GSEA (Broad Institute, http://www.broadinstitute.org/gsea/msigdb/index.jsp).

## NO release detection in eMSCs

For in situ visualizations of intracellular NO, a specific membrane-permeable NO molecular probe, DAF-FM diacetate, was used, which was loaded into cells and produced DAF-FM by intracellular esterase (*Hou et al., 2019*). Cultured eMSCs were incubated with 5 μM DAF-FM diacetate at 37°C for 20 min followed by extensive washing and then incubated with 5 μg/mL NO prodrug at 37°C for 6 hr, according to the manufacturer's instructions (S0019; Beyotime).

## Assessment of NO release in vivo by chemiluminescence

Chemiluminescence methods were adopted to measure NO release from tissues by using a nitric oxide analyzer (NOA 280i, Zysense, USA). On day 7 after injury, the mice were sacrificed, and the

samples were harvested. Then, the tissues were homogenized, and the supernatant was collected for NO detection. The samples were immersed in 5 mL of vanadium (III) chloride (50 mM), and the generated NO gas was diffused in the test solution and transported to the NO analyzer by a stream of $N_2$. The generated NO was calculated using $NaNO_3$ as a standard.

## Mice and animal model

C57BL/6 albino mice (10 weeks of age, female and male) were purchased from Charles River Laboratories (Charles River). The I/R injury-induced AKI model was established as previously described (*Zhang et al., 2020b*). Briefly, mice were anesthetized by intraperitoneal injection of 2.5% avertin (240 mg/kg, Sigma-Aldrich, Oakville, ON, Canada). A longitudinal incision was made on the back of the mouse above the left kidney to expose the kidney. Clamping the left renal artery and vein with an arterial clamp for 45 min resulted in ischemic blackening of the kidney and visually verified reperfusion before closing the incision. To avoid the compensatory effect of the contralateral kidney, the right kidney was dissected. After reperfusion, $1\times10^6$ cells (30 µL) of eMSCs were transplanted into the left kidney through renal parenchymal injection. Subsequently, 200 µL of 1 mg/mL NO prodrug, MGP, was administered via tail vein injection every other day until the end of the experiment. To monitor angiogenesis in real time in vivo, C57BL/6 albino and outbred (Nu/Nu) background Vegfr2-luciferase transgenic mice (10 weeks of age, female) were used to establish an AKI model induced by I/R injury (*Zhang et al., 2023*; *Huang et al., 2022*). The treatment of animals and the experimental procedures of the present study adhered to the Nankai University Animal Care and Use Committee Guidelines that conform to the Guidelines for Animal Care approved by the National Institutes of Health (NIH).

## NO release in the supernatant of cell culture

NO production in culture media and tissues was measured using the Nitric Oxide Assay Kit (S0021; Beyotime) based on the Griess reaction according to the manufacturer's instructions. For in vitro NO release, $5\times10^3$ eMSCs were seeded in 24-well plates and then incubated with different concentrations (0, 2, 10 µg/mL, 1 mL/well) of the NO-prodrug MGP. At each predetermined time interval, 50 µL of the culture media was transferred to a 96-well plate, and 50 µL of Griess I and 50 µL of Griess II were added in sequence. Finally, a microplate reader (Promega) was used to measure the optical density values of the samples at 540 nm.

## Measurement of NO release in vitro by the EPR technique

Intracellular NO production was detected by EPR. In brief, sodium DETC (9 mg) and $FeSO_4\bullet7H_2O$ (5.6 mg) were dissolved in 20 mL of deoxygenated Krebs/HEPES solution. Equal volumes of $FeSO_4\bullet7H_2O$ solutions were rapidly added into sodium DETC to form a 0.5 mM Fe•(DETC)$_2$ colloid solution. eMSCs were rinsed with modified Krebs/HEPES buffer and incubated with freshly prepared NO•-specific spin trap Fe•(DETC)$_2$ colloid (0.5 mM). Meanwhile, 10 µg/mL NO prodrug MGP was added to the buffer for 1 hr. Gently collected cell suspensions were snap-frozen in liquid nitrogen. Then, the cells were ultrasonically broken and extracted by ethyl acetate (50 µL), and the EPR was measured at room temperature. The following acquisition parameters were used: modulation frequency, 100 kHz; microwave power, 10 mW; modulation amplitude, 2 G; and number of scans, 15. The double-integrated area of the EPR spectra was calibrated into concentrations of DETC$_2$-Fe-NO using TEMPO as a standard.

## Measurement of NO release in vivo by the EPR technique

The NO production of tissues in vivo was measured by EPR spectroscopy as previously described (*Zhang et al., 2020a*). C57BL/6 albino mice (20–25 g) were anesthetized using $1\times10^6$ cells (30 µL) of eMSCs transplanted into the left kidney through renal parenchymal injection. Then, DETC sodium salt (500 mg/kg, Sigma-Aldrich) dissolved in water (250 mM) was injected subcutaneously. After 5 min, ammonium ferrous sulfate (50 mM) dissolved in citrate solution (250 mM) was administered by subcutaneous injection (2 mL/kg). Subsequently, 200 µL of 1 mg/mL MGP was administered via tail vein injection and allowed to circulate for 1 hr. Tissues were harvested and quickly frozen in liquid nitrogen. After that, the samples were crumbled into small pieces and extracted with 200 µL of ethyl acetate immediately. X-band EPR was used to measure the ethyl acetate extract in 50 µL capillary tubes at room temperature. The instrument settings were as follows: modulation frequency, 100 kHz; microwave power, 10 mW; modulation amplitude, 2 G; the number of scans, 30. The double-integrated

area of the EPR spectrum was calibrated into concentrations of $NO\text{-}Fe^{2+}(DETC)_2$ using TEMPO as a standard.

## In vitro proangiogenic assay

To analyze the proangiogenic effect of eMSCs in vitro, a scratch wound-healing assay was performed to evaluate the migration capacity of HUVECs using a coculture system. HUVECs were seeded in a six-well plate cocultured with eMSCs or eMSCs with MGP administration on a 24 mm Transwell (3412, Corning). When HUVECs reached confluence, scratch wounds were generated using the tip of a 10 μL micropipette. Images of five microscopic fields per well were taken at 0 hr and 12 hr, and the percentage of wound healing was quantitated using ImageJ software.

For the tube formation assay, 150 μL/well growth factor-reduced cold Matrigel (Corning, Corning, NY, USA) was coated onto 48-well plates and then incubated at 37°C for 30 min for gelatinization. Then, $3\times10^4$ HUVECs were seeded per well and cocultured with eMSCs or eMSCs with MGP administration in a 6.5 mm Transwell (3413, Corning). After incubation for 12 hr, the number of branches and nodes was measured in three randomly selected microscopic fields using ImageJ software as described before.

## Gene expression analysis

Total RNA was isolated from kidney tissues or cells using TRIzol reagent (Invitrogen) according to the manufacturer's instructions and spectrophotometrically quantified using a NanoDrop One (Thermo, USA), and RNA integrity was tested by 1% agarose gel electrophoresis. For RNA reverse transcription into complementary DNA (cDNA), 1.0 μg of total RNA was treated with the BioScript All-in-One cDNA Synthesis SuperMix kit (Bimake, Houston, TX, USA). RT-quantitative polymerase chain reaction (qPCR) analysis was performed using a CFX96TM Real-Time System (Bio-Rad, Hercules, CA, USA) in a 20 μL reaction volume containing qPCR SYBR green master mix (Yeasen, Shanghai, China). The $2^{-\Delta\Delta Ct}$ method was used to determine the relative mRNA folding changes. Primers are listed in *Supplementary files 2 and 3*.

## Immunostaining analysis

For immunofluorescence staining, 8 μm cryosections were returned to room temperature for 30 min and then washed three times in PBS. Next, the cells were incubated with 5% BSA containing 0.1% Triton X-100 at room temperature for 1 hr and then incubated with a series of primary antibodies overnight at 4°C. The following secondary antibodies were used: Alexa Fluor 488-, Alexa Fluor 594-, Alexa Fluor 555-, and Alexa Fluor 647-labeled goat anti-rabbit, goat anti-mouse, or goat anti-rat (1:400; Invitrogen). FITC-labeled Lotus tetragonolobus lectin (LTL, 1:400, Vector Laboratories, Burlingame, CA, USA) was used to stain proximal tubules. DAPI was used for nuclear counterstaining. For quantification, the positive immunostaining area or positively stained cells were measured by ImageJ software. For immunohistochemistry staining of kidney tissues, after dewaxing and rehydration, paraffin sections were incubated with primary antibodies overnight at 4°C. Horseradish peroxidase-conjugated secondary antibody was used (1:200, SP-9000, ZSGB-BIO, Beijing, China), and a DAB staining kit was used to develop colors (ZLI-9017, ZSGB-BIO). The integrated optical density values of positive staining were measured using Image-pro plus 6.0 software. Antibodies are listed in *Supplementary file 1*.

## Renal oxidative stress-related index

Kidney samples (100 mg) were collected and homogenized in 900 μL of cold PBS with a tissue homogenizer to prepare a 10% tissue homogenate. Then, the activities of the oxidative stress-related factors MDA, SOD, and GSH/GSSG were detected with a lipid peroxidation MDA Assay Kit (A003-1-2, Nanjing Jiancheng Bioengineering Institute), SOD Assay Kit (A001-3-2, Nanjing Jiancheng Bioengineering Institute), and GSH and GSSG Assay Kit (S0053, Beyotime).

## Statistical analysis

To check for significant differences, one-way analysis of variance (ANOVA) was used followed by Tukey's HSD test when needed to compare three or more experimental groups of data, and Student's t-test was used to compare two groups of data. All data were analyzed using SPSS Statistics, version

25.0 (IBM SPSS Inc, Chicago, IL, USA). All quantification data are presented as the mean ± SD of at least three independent replicates.

## Acknowledgements

This study was financially supported by the National Key R&D Program of China (2017YFA0103200), the National Natural Science Foundation of China (81925021, U2004126, 82330066), the Tianjin Natural Science Foundation (22JCZXJC00170, 21JCZDJC00070), the Open funding from Nankai University Eye Institute (NKYKD202203), and the Tianjin Key Medical Discipline (Specialty) Construction Project (TJYXZDXK-043A).

## Additional information

### Competing interests

Zhibo Han: is an employee of AmCellGene Co., Ltd. The author has no further interests to declare. Zhong-Chao Han: is an employee of Health & Biotech Co and AmCellGene Co., Ltd. The author has no further interests to declare. The other authors declare that no competing interests exist.

### Funding

| Funder | Grant reference number | Author |
|---|---|---|
| National Key Research and Development Program of China | 2017YFA0103200 | Zongjin Li |
| National Natural Science Foundation of China | U2004126 | Zongjin Li |
| National Natural Science Foundation of China | 81925021 | Qiang Zhao |
| Tianjin Natural Science Foundation | 22JCZXJC00170 | Zongjin Li |
| Open Funding from Nankai University Eye Institute | NKYKD202203 | Zongjin Li |
| Tianjin Natural Science Foundation | 21JCZDJC00070 | Zongjin Li |
| National Natural Science Foundation of China | 82330066 | Qiang Zhao |
| Tianjin Key Medical Discipline (Specialty) Construction Project | TJYXZDXK-043A | Zongjin Li |

The funders had no role in study design, data collection and interpretation, or the decision to submit the work for publication.

### Author contributions

Haoyan Huang, Data curation, Formal analysis, Investigation, Methodology, Writing – original draft; Meng Qian, Huifang Li, Validation, Investigation, Methodology; Yue Liu, Data curation, Investigation, Methodology, Writing – original draft; Shang Chen, Formal analysis, Validation, Visualization; Zhibo Han, Resources, Validation, Investigation; Zhong-Chao Han, Conceptualization, Resources, Supervision, Project administration; Xiang-Mei Chen, Conceptualization, Resources, Supervision, Funding acquisition, Project administration; Qiang Zhao, Conceptualization, Supervision, Funding acquisition, Methodology, Project administration, Writing – review and editing; Zongjin Li, Data curation, Funding acquisition, Investigation, Methodology, Writing – original draft, Writing – review and editing

### Author ORCIDs

Haoyan Huang http://orcid.org/0000-0001-7753-7242

Zongjin Li  https://orcid.org/0000-0002-4603-3743

## Ethics

This study was performed in strict accordance with the recommendations in the Guide for the Care and Use of Laboratory Animals of the National Institutes of Health. All of the animals were handled according to the Nankai University Animal Care and Use Committee Guidelines (approval no. 2021-SYDWLL-000426).

### Decision letter and Author response
Decision letter https://doi.org/10.7554/eLife.84820.sa1
Author response https://doi.org/10.7554/eLife.84820.sa2

## Additional files

### Supplementary files
• Supplementary file 1. Antibodies used for western blotting and immunostaining analysis.

• Supplementary file 2. List of human primer sequences used for quantitative polymerase chain reaction (qPCR) analysis in this study.

• Supplementary file 3. List of mouse primer sequences used for quantitative polymerase chain reaction (qPCR) analysis in this study.

• MDAR checklist

### Data availability
All raw data for bulk RNA sequencing has been deposited to NCBI (BioProject PRJNA910491); Source data file has been provided for *Figure 1—source data 1*.

The following dataset was generated:

| Author(s) | Year | Dataset title | Dataset URL | Database and Identifier |
|---|---|---|---|---|
| Li Z | 2023 | Genetically engineered mesenchymal stem cells as a nitric oxide reservoir for acute kidney injury therapy | https://www.ncbi.nlm.nih.gov/bioproject/?term=PRJNA910491 | NCBI BioProject, PRJNA910491 |

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
