## [Editor Report]

The study provides compelling evidence that treatment with the newly designed NO prodrug, MGP, selectively triggers NO release from your genetically engineered MSCs.

The significance of the study is that it provides in vivo demonstration that MSCs can release NO in a spatiotemporal manner in a mouse model of acute kidney injury thus contributing to regeneration. This constitutes a landmark finding with profound implications that are expected to have widespread influence. The work not only shows the therapeutic efficiency of MSCs, but also holds promises for regenerative therapy by enhancing the therapeutic efficiency of stem cells. Thus, it is felt that the newly generated tools will be used by many investigators thus making the findings interesting to a broad audience.

---

## [Decision Letter]

**Decision letter after peer review:**

Thank you for submitting your article "Genetically engineered mesenchymal stem cells as a nitric oxide reservoir for acute kidney injury therapy" for consideration by *eLife*. Your article has been reviewed by 3 peer reviewers, and the evaluation has been overseen by a Reviewing Editor and Martin Pollak as the Senior Editor. The following individual involved in the review of your submission has agreed to reveal their identity: Anton Jan van Zonneveld (Reviewer #3).

The authors generated engineered MSC (eMSC) to produce mutant b-GAL^H363A^, and when stimulated with a pro-drug (MGP) they can release NO. These cells were tested in vivo in a mouse model of AKI. When MGP is systemically administrated in AKI mice, it can induce eMSC to release NO in a precise and spatiotemporal manner, possibly enhancing the therapeutic efficacy of these stem cells.

The authors have conducted a very interesting study. They provide proof of concept that the pharmacokinetically controlled generation of NO in transplanted mesenchymal stem cells can augment the regenerative actions of these cells in ischemia-reperfusion injury of the kidney. This could impact the treatment of patients with acute kidney injury. State-of-the-art techniques are used to confirm the NO generation and, the data are consistent and very convincing. Thus, these results are likely of interest to the renal scientific community, especially in the context of acute kidney injury.

Essential revisions:

1) A criticism relevant to the clinical arena is the lack of information on the survival of the MSC after transplantation and the introduction of novel antigens associated with the engineered cells.

2) The mice utilized were C57/Bl6 females, the most resistant sex and strain to both acute and chronic kidney injury.

3) The use of human placental MSCs only; as such it is not clear whether other MSCs will perform equally well.

4) In general, the methods should contain more details on experimental procedures, nature, and isolation of cells.

5) The paper should contain more information on biological replicates and groups, time points, deep sequencing, and QC for the bulk RNA-seq should be provided.

6) Some of the figures are either not convincing or difficult to understand and a more detailed figure legend could help in appreciating the results presented.

Based on these criticisms, the authors should:

1) Provide evidence that other MSC can equally perform in vivo;

2) Perform AKI studies on male mice which are more prompt to injury thus facilitating the evaluation of the beneficial effects of MSCs;

3) Extensively re-write methods and figure legend and provide more convincing figures.

*Reviewer #1 (Recommendations for the authors):*

It is key to better explain the important assays in the text and figure legends.

1. Please describe the AKI model used on Page 9, in the figure 3 legend, etc. so that the reader knows which AKI model is being utilized immediately. The AKI models available are quite variable so this is important. Why were C57/Bl6 females, the strain and sex most resistant to kidney injury, chosen for this study? More details need to be provided on page 19 as well. Yes, it says "previously described" but as this is a key detail, the authors should give a description in this methods section of this paper.

2. Please give the ratio of the GSH/GSSG ratio (reduced to oxidized glutathione) rather than just GSH. To determine oxidative stress, the ratio is needed.

3. In Figure S4, "NO release in PBS with β-GAL^H363A^" the Griess assay was provided but there are no controls.

4. Did the authors try MSCs from mice vs human, or from other sources vs placenta? The human placenta source of the MSCs should be sporadically highlighted throughout including in the paper abstract, not only mentioned once in the methods section, as the conclusions of this manuscript could possibly be specific to human placental MSCs unless others were tested. Also, the authors should add sentences on why there was a lack of immune rejection of the MSCs in this study.

*Reviewer #2 (Recommendations for the authors):*

See comments below for improvements.

– Methods are lacking in important details. How were the cells isolated (it can not be referenced to a previous publication, details should be also included in this Manuscript)? How many samples were tested (biological replicates)? How many cells are used for each experiment? Also, a lot more information (groups, time points, biological replicates, deep sequencing, QC for the bulk RNA-seq should be provided). Were human cells injected into non-immunodeficient mice? How was the AKI induced? How many mice per group? It is not clear in which experiments the HK-2 and the HEK-293T cells were used.

– Figure 1C. It is not clear the percentage of eMSC that are positive for both b-GAL as well as for RFP. The first panel does not seem to be the same as the second one, therefore it is hard to understand the merged panel. Dapi should have been used to distinguish the nuclei. A Flow should have been provided to determine the percentage of cells that are positive for both markers. In this case, the WB is not sufficient, a qualitative and quantitative analysis is important.

– Please eliminate this sentence " eMSCs showed similar morphology and expression

patterns, indicating minimal side effects of genetic modification (Figure S2)." Similar morphology does not indicate minimal side effects of genetic modification. Even if the morphology is similar, the gene expression could also be very different. Please eliminate this sentence.

– Figure S5 is very confusing. In the Results section, it seems that immunostaining for the expression of proliferation-related genes is shown. There are no staining and usually, staining does not show gene expression (unless it is in situ hybridization).

– Time points for the experiments are not always clear. For example, Figure S6 and S7 have different time points (48hr vs 24hr or 12hr). It is very hard to understand the rationale behind these choices.

– It is supposed that Figure S8 is showing bulk RNA-seq? please specify. Legends are not clear. Is red showing upregulated genes? Are the data significant? How many cells were sequenced and how many biological replicates? Same questions for Figure 2.

– "Together, these results demonstrated that NO might be able to ameliorate the

oxidative stress of eMSCs, thus elongating the survival of eMSCs" This is a very strong statement that is not supported by the results presented. Please revise.

– Figure 3. It is hard to understand the time points for the experiments shown in this Figure. Are they all after 1 week after administration of the cells? How the cells were administered? Figure 3H is not indicative. Are the sham mice WT mice with no damage? Based on the images, the last panel in Figure 3H seems to show that eMSC+MGP is worse than the eMSC; the eMSC+MGP seems to have more cast formation and worse histology.

– Figure 5: Why a time point of 1 week was chosen? What happens at a later stage? Figure 5A: how do the authors explain the signaling in the sham and in the PBS groups?

– Figure S13A: it is understood that the treatment with MSC should lower the lipids deposition. It is strange though that the sham group does not have any lipid deposition (none).

This is an interesting work but the Authors should provide a better method and legend description as well as focus the discussion on their findings, and correct the figures as suggested thus providing more evidence of the effect of eMSC.

*Reviewer #3 (Recommendations for the authors):*

The abstract starts with the notion that "the immunogenicity and long-term toxicity of artificial carriers hinders the potential clinical translation of this gas therapeutics…" One could argue that, although MSC has been shown to display immunomodulatory effects, MSC from the allogenic origin, as well as the mutant β-galactosidase, may provide novel antigens that can be targeted by the host's immune system. It would be good to discuss this issue.

The authors report in Figure S4 and 1F that the addition of MGP to the MSC elicits a release of NO based on a Griess assay (S4) and staining of DAF-FM. This is a nice result, but it has to be taken into account that Griess and DAF-DA also detect NO indirectly via oxidation products. Therefore, a biological sensor of NO production in the MSC could be the generation of cGMP. cGMP can be easily measured and is, to this reviewer's knowledge, the most likely effector of the effects of NO on MSC. Alternatively, and surprisingly the authors do use a NO analyzer and even EPR! in the in vitro studies, why not use these also for the validation of NO release by the cells in vitro?

Figures S6 and S7 demonstrate the modest effect of the MGP-dependent NO generation on H2O2-induced apoptosis and survival. A missing control is the effect of MGP on H2O2-induced apoptosis and survival of the control cells. May spontaneous release of NO, without the help of the galactosidase is enough for the effect?

For the in vivo studies described in figure 3 it would be helpful I a description in the methods section would be given on how the MSC are transplanted.

The model of IR can be performed in different ways and even though the authors refer to a published paper it should be stated whether this is a unilateral model or a bilateral model and whether or not the contralateral right kidney is dissected (most likely).

In the angiogenesis studies it is surprising that PBS stimulates angiogenesis, can this be discussed? Again, the MGP-only control is lacking. Nice this is confirmed in HUVEC. Do you have any indication of how long the MSC survives after transplantation?

---

## [Author Response]

Essential revisions:1) A criticism relevant to the clinical arena is the lack of information on the survival of the MSC after transplantation and the introduction of novel antigens associated with the engineered cells.

Thank you very much for your comments. In the present study, MSCs were genetically modified with triple reporter genes, mutant β-galactosidase (β-GAL^H363A^), Rluc, and RFP. Engineered mesenchymal stem cells (eMSCs) can produce mutant β-galactosidase and trigger the release of nitric oxide (NO) when the NO prodrug is administered systemically. Furthermore, Rluc can be used for in vivo bioluminescence imaging of the survival of MSCs after transplantation, and RFP can be used for immunofluorescence staining. The Rluc imaging of eMSCs, as reflected by BLI signals, showed that 1×10^6^ eMSCs could be retained in the I/R injured kidney for 5 days, while eMSCs with MGP administration could significantly improve cell survival, indicating that NO could significantly promote cell survival in vivo. Moreover, RFP expression of eMSCs was also confirmed in the kidney, and higher cell proliferation and cell retention were revealed with MGP administration. These data can be found in Figures 3A-C of the revised manuscript.

MSCs have long been reported to be hypoimmunogenic or immune privileged; however, several studies have reported the generation of antibodies against MSCs, especially in genetically modified MSCs, such as MSCs transfected with erythropoietin (*Mol. Ther. 17, 369–372 (2009)*). Here, triple reporter genes were applied. In future clinical applications, no therapeutic function genes, Rluc and RFP, should be removed to minimize immune rejection of MSCs.

Previous studies revealed that MSCs do not express blood-group antigens or MHC class II antigens (*J Am Soc Nephrol. 25(5), 877-83 (2014)*), and the triple reporter genes, mutant β-galactosidase (β-GAL^H363A^), Rluc and RFP, do not express themselves on the cell membrane, suggesting that there is no rapid immune rejection of genetically modified MSCs. MSCs may exert therapeutic function through a brief 'hit and run' mechanism, protecting MSCs from immune detection and prolonging their persistence in vivo may improve clinical outcomes and prevent patient sensitization toward donor antigens (*Nature Biotechnology 32, 252–260 (2014)).*It is widely acknowledged that MSCs are highly immunomodulatory (*Cellular and Molecular Immunology 20, 555–557 (2023)*). The rejection of MSCs depends on the balance between the expression of immunogenic and immunosuppressive factors by MSCs (*Nature Biotechnology 32, 252–260 (2014)*)*.* Strategies to increase eMSC survival after transplantation will be further investigated in our future study.

2) The mice utilized were C57/Bl6 females, the most resistant sex and strain to both acute and chronic kidney injury.

Thank you very much for your comments. Although male mice are more susceptible to kidney injury from ischemia‒reperfusion, evidence suggests that an ischemia time of 34 minutes in a female murine model resulted in an injury comparable to 22 minutes for males (*J Am Soc Nephrol. 33(2), 279–289 (2022)*). In our work, we optimized the clamping time of the renal pedicle for 45 minutes to generate an equivalent injury in female C57BL6/J mice compared to their male counterparts. Furthermore, the degree of kidney injury was assessed by serum creatinine, urea nitrogen, renal histopathology, and acute kidney injury marker (Kim-1) in our work, indicating that the AKI model was established successfully in female mice (Figure 3D-H of previous version). As suggested, we conducted additional experiments to investigate the therapeutic effect of eMSCs with MGP administration in male mice. We measured serum creatinine and urea nitrogen levels, renal histopathology, and the acute kidney injury marker Kim-1 to evaluate the therapeutic potential of eMSCs with MGP administration. Similar results were observed in male mice. Data from male mice were combined with data from female mice, which can be found in the revised Figure 3D-H and Figure 5A, C, E.

3) The use of human placental MSCs only; as such it is not clear whether other MSCs will perform equally well.

We are so grateful for your meaningful comments and suggestions. As suggested, we conducted additional experiments to investigate whether other MSCs will perform equally well. Human adipose-derived MSC (AD-MSC), human umbilical cord MSC (hUC-MSC) were genetically modified and our results confirmed that AD-MSC and hUC-MSC perform well with this system. We first performed an EPR technique to directly measure the NO levels of eAD-MSC and ehUC-MSC with MGP administration in vitro and in vivo. The therapeutic effects of the NO-eAD-MSC and NO-ehUC-MSC system in AKI mice were then evaluated by histological analysis and renal function analysis. An obvious characteristic triplet EPR signal (NO signal) was observed in eAD-MSC and ehUC-MSC with the administration of MGP in vitro and in vivo. Furthermore, the eAD-MSC-NO system showed significantly lower levels of SCr and BUN. Furthermore, tubular dilation, cast formation, and massive loss of brush borders at the initial stage after kidney injury (3 d) were obviously reduced after treatment with the NO-eMSC system. Furthermore, KIM-1 expression was significantly decreased in the proximal tubules of the kidneys in the eAD-MSC with MGP administration group. Thus, our eMSC and NO prodrug system provides a tunable platform for NO delivery and holds promise for regenerative therapy. Data are shown in Figure 5—figure supplement 1-3.

4) In general, the methods should contain more details on experimental procedures, nature, and isolation of cells.

We are grateful for your suggestion. As suggested, we have carefully revised our manuscript and supplemented the experimental details in the “Materials and methods” section. Therefore, a detailed amendment can be found in the revised manuscript.

5) The paper should contain more information on biological replicates and groups, time points, deep sequencing, and QC for the bulk RNA-seq should be provided.

Thank you for your comments and advice. We have carefully revised our manuscript and supplemented the experimental details in the revised manuscript. Briefly, engineered MSCs (eMSCs) treated with MGP (2 μg/mL) or PBS for 24 h were collected. For each group, 3 duplicates were prepared for RNA-seq analysis. Total RNA from eMSCs and eMSCs with MGP administration was extracted using TRIzol reagent (Invitrogen, USA). The amount and purity of RNA in each sample was quantified using a NanoDrop ND-1000 (NanoDrop, Wilmington, DE, USA). The integrity of the RNA was demonstrated with a Bioanalyzer 2100 (Agilent, CA, USA), RIN > 7.0, and confirmed by denaturing agarose gel electrophoresis. Total RNA was sent for RNA-Seq library construction and high-throughput sequencing on an Illumina Novaseq 6000 with 2×150 bp paired-end sequencing (PE150). Raw reads obtained from RNA-Seq were preprocessed. The reads that contained adapter contamination were removed using Cutadapt software. After removing low-quality and undetermined bases, we mapped the reads to the genome using HISAT2 software. The results of quality control (QC) of the RNA-seq profiles of eMSCs and eMSCs with MGP administration are shown in Figure 2—figure supplement 1.

6) Some of the figures are either not convincing or difficult to understand and a more detailed figure legend could help in appreciating the results presented.

We apologize for our unclear descriptions. As suggested, we extensively rewrote all figure legends in the revised manuscript.

Based on these criticisms, the authors should:1) Provide evidence that other MSC can equally perform in vivo;

Human adipose-derived MSC (ADSC) and human umbilical cord MSC (hUC-MSC) were tested in vivo and results similar to those of hP-MSC were observed.

2) Perform AKI studies on male mice which are more prompt to injury thus facilitating the evaluation of the beneficial effects of MSCs;

As suggested, we conducted additional experiments to investigate the therapeutic effect of eMSCs with MGP administration in male mice.

3) Extensively re-write methods and figure legend and provide more convincing figures.

The method and figure legend sections were rewritten, and Figure 1C was replaced.

Reviewer #1 (Recommendations for the authors):It is key to better explain the important assays in the text and figure legends.

We are grateful for your suggestion. As suggested, we have carefully revised our manuscript and supplemental documents and explained important assays in more detail.

1. Please describe the AKI model used on Page 9, in the figure 3 legend, etc. so that the reader knows which AKI model is being utilized immediately. The AKI models available are quite variable so this is important. Why were C57/Bl6 females, the strain and sex most resistant to kidney injury, chosen for this study? More details need to be provided on page 19 as well. Yes, it says "previously described" but as this is a key detail, the authors should give a description in this methods section of this paper.

Thank you for your kind comments and suggestions. In our study, renal ischemia/reperfusion injury models were established to mimic acute kidney injury. Although male mice are more susceptible to kidney injury from ischemia‒reperfusion, evidence suggests that an ischemia time of 34 minutes in a female murine model resulted in an injury comparable to 22 minutes for males (*J Am Soc Nephrol. 33(2), 279–289 (2022)*). In our work, we optimized the clamping time of the renal pedicle for 45 minutes to generate an equivalent injury in female C57BL6/J mice compared to their male counterparts. As suggested, we conducted additional experiments to investigate the therapeutic effect of eMSCs with MGP administration in male mice. We measured serum creatinine and urea nitrogen levels, renal histopathology, and the acute kidney injury marker Kim-1 to evaluate the therapeutic potential of eMSCs with MGP administration. Our results demonstrated that eMSCs with MGP administration could significantly improve renal function by decreasing serum creatinine and urea nitrogen levels. IRI caused the formation of a necrotic tubule mass accompanied by cast formation and loss of the brush border, and this process was significantly reduced by eMSC with MGP administration. Immunostaining of the kidney injury marker (Kim-1) revealed similar results. Data from male mice were combined with data from female mice, which can be found in the revised Figure 3D-H and Figure 5A-E.

We have described the AKI model in more detail in the Results, Methods, and Figure Legend sections.

“Briefly, mice were anesthetized by intraperitoneal injection of 2.5% avertin (240 mg/kg, Sigma‒Aldrich, Oakville, ON, Canada), and protective ophthalmic ointment was applied to mouse eyeballs to maintain moisture. At the same time, remove the hair from the surgical site on the back of the mouse with a hair scraper and depilatory cream and disinfect it with povidone-iodine. A longitudinal incision was made on the back of the mouse above the left kidney to expose the kidney. Using ophthalmic forceps, dissect bluntly the fat and connective tissue surrounding the renal arteries and veins to expose the renal arteries and veins. Clamping of the renal artery and vein with an arterial clamp for 45 min resulted in ischemic blackening of the kidney and visually verified reperfusion.”

2. Please give the ratio of the GSH/GSSG ratio (reduced to oxidized glutathione) rather than just GSH. To determine oxidative stress, the ratio is needed.

We appreciate your advice and examined the GSH/GSSG ratio to determine oxidative stress and added this result in the revised manuscript. Both GSH and the GSH/GSSG ratio were significantly higher and GSSG was lower in the kidneys of eMSCs treated with MGP than in the injured kidneys treated with PBS. The data are shown in Figure 7—figure supplement 1E.

3. In Figure S4, "NO release in PBS with β-GAL^H363A^" the Griess assay was provided but there are no controls.

Thank you for your comments. We provide control of the Greiss assay now**.**

4. Did the authors try MSCs from mice vs human, or from other sources vs placenta? The human placenta source of the MSCs should be sporadically highlighted throughout including in the paper abstract, not only mentioned once in the methods section, as the conclusions of this manuscript could possibly be specific to human placental MSCs unless others were tested. Also, the authors should add sentences on why there was a lack of immune rejection of the MSCs in this study.

We are grateful for your meaningful comments and suggestions. As suggested, we conducted additional experiments to investigate whether other MSCs will perform equally well. Human adipose-derived MSC (AD-MSC), human umbilical cord MSC (hUC-MSC) were genetically modified and our results confirmed that AD-MSC and hUC-MSC perform well with this system. We first performed an EPR technique to directly measure the NO levels of eADSC and eUC with MGP administration in vitro and in vivo. The therapeutic effects of the NO-eADSC and NO-eUC system in AKI mice were then evaluated by histological analysis and renal function analysis. An obvious characteristic triplet EPR signal (NO signal) was observed in eAD-MSC and ehUC-MSC with the administration of MGP in vitro and in vivo. Furthermore, the NO-eAD-MSC and NO-ehUC-MSC systems both showed significantly lower levels of SCr and BUN. Furthermore, tubular dilation, cast formation, and massive loss of brush borders at the initial stage after kidney injury (3 d) were obviously reduced after treatment with the NO-eADSC and NO-eUC system. Furthermore, KIM-1 expression was significantly decreased in the proximal tubules of the kidneys in the eADSC and eUC with MGP administration groups. Thus, our eMSC and NO prodrug system provides a tunable platform for NO delivery and holds promise for regenerative therapy. Data are shown in Figure 5—figure supplement 1-3.

Regarding the issue of new antigens associated with engineered cells, MSCs have long been reported to be hypoimmunogenic or immune privileged; however, several studies have reported the generation of antibodies against MSCs, especially in genetically modified MSCs, such as MSCs transfected with erythropoietin (*Mol. Ther. 17, 369–372 (2009)*). Here, triple reporter genes were applied. In future clinical applications, no therapeutic function genes, Rluc and RFP, should be removed to minimize the immune rejection of MSCs.

Previous studies revealed that MSCs do not express blood-group antigens or MHC class II antigens (*J Am Soc Nephrol. 25(5), 877-83 (2014)*), and the triple reporter genes, mutant β-galactosidase (β-GAL^H363A^), Rluc and RFP, do not express themselves on the cell membrane, suggesting that there is no rapid immune rejection of genetically modified MSCs. MSCs may exert therapeutic function through a brief 'hit and run' mechanism, protecting MSCs from immune detection and prolonging their persistence in vivo may improve clinical outcomes and prevent patient sensitization toward donor antigens (*Nature Biotechnology 32, 252–260 (2014)).*It is widely acknowledged that MSCs are highly immunomodulatory (*Cellular and Molecular Immunology 20, 555–557 (2023)*). The rejection of MSCs depends on the balance between the expression of immunogenic and immunosuppressive factors by MSCs (*Nature Biotechnology 32, 252–260 (2014)*)*.* Strategies to increase eMSC persistence should be further investigated in future studies.

Reviewer #2 (Recommendations for the authors):See comments below for improvements.– Methods are lacking in important details. How were the cells isolated (it can not be referenced to a previous publication, details should be also included in this Manuscript)? How many samples were tested (biological replicates)? How many cells are used for each experiment? Also, a lot more information (groups, time points, biological replicates, deep sequencing, QC for the bulk RNA-seq should be provided). Were human cells injected into non-immunodeficient mice? How was the AKI induced? How many mice per group? It is not clear in which experiments the HK-2 and the HEK-293T cells were used.

Thank you for your comments. More experimental details have been supplemented in the 'Methods' and 'Figure legends' sections.

– Figure 1C. It is not clear the percentage of eMSC that are positive for both b-GAL as well as for RFP. The first panel does not seem to be the same as the second one, therefore it is hard to understand the merged panel. Dapi should have been used to distinguish the nuclei. A Flow should have been provided to determine the percentage of cells that are positive for both markers. In this case, the WB is not sufficient, a qualitative and quantitative analysis is important.

We conducted FACS analysis, and the data revealed that positivity for both RFP and β-GAL^H363A^ was approximately 50% (Figure 1—figure supplement 1A). This population of double-positive cells was sorted for further experiments. Figure 1C was replaced for a better description of both RFP and β-GAL^H363A^ expression in eMSCs.

– Please eliminate this sentence " eMSCs showed similar morphology and expressionpatterns, indicating minimal side effects of genetic modification (Figure S2)." Similar morphology does not indicate minimal side effects of genetic modification. Even if the morphology is similar, the gene expression could also be very different. Please eliminate this sentence.

We appreciate your advice and have deleted this sentence.

– Figure S5 is very confusing. In the Results section, it seems that immunostaining for the expression of proliferation-related genes is shown. There are no staining and usually, staining does not show gene expression (unless it is in situ hybridization).

This has been corrected.

– Time points for the experiments are not always clear. For example, Figure S6 and S7 have different time points (48hr vs 24hr or 12hr). It is very hard to understand the rationale behind these choices.

Thank you for your comments, and we apologize for our misleading description. Considering that the cell-protective effects of a low NO concentration can be explained by Figure 1—figure supplement 5, we deleted Figure S6 of previous version.

– It is supposed that Figure S8 is showing bulk RNA-seq? please specify. Legends are not clear. Is red showing upregulated genes? Are the data significant? How many cells were sequenced and how many biological replicates? Same questions for Figure 2.

Figures 2 and S8 show bulk RNA-seq. We cleared this.

– "Together, these results demonstrated that NO might be able to ameliorate theoxidative stress of eMSCs, thus elongating the survival of eMSCs" This is a very strong statement that is not supported by the results presented. Please revise.

We appreciate your advice and have rewritten this sentence during revision.

– Figure 3. It is hard to understand the time points for the experiments shown in this Figure. Are they all after 1 week after administration of the cells? How the cells were administered? Figure 3H is not indicative. Are the sham mice WT mice with no damage? Based on the images, the last panel in Figure 3H seems to show that eMSC+MGP is worse than the eMSC; the eMSC+MGP seems to have more cast formation and worse histology.

Thank you for your comments. Figure 3A, BLI analysis was performed after cell transplantation from day 1. In our work, 1×10^6^ eMSCs (30 μL, 3 sites) were transplanted into the left kidney by renal parenchymal injection. Figure 3C-H, all are on day 3 after administration of the cells. Furthermore, we replaced the representative picture in Figure 3H as suggested.

– Figure 5: Why a time point of 1 week was chosen? What happens at a later stage? Figure 5A: how do the authors explain the signaling in the sham and in the PBS groups?

According to Figure 5A, peak BLI signals were detected at week 1. Therefore, the time point for angiogenic analysis of CD31 staining was selected. Vegfr2-Fluc mice, transgenic mice expressing firefly luciferase (Fluc) under the promoter of vascular endothelial growth factor receptor 2 (VEGFR2-Luc), were used here. Injury in the sham and PBS groups also initiated angiogenesis during tissue regeneration, while BLI signals were significantly enhanced in the eMSC with MGP administration group.

– Figure S13A: it is understood that the treatment with MSC should lower the lipids deposition. It is strange though that the sham group does not have any lipid deposition (none).

Accumulation of lipids in tubular epithelial cells has been proposed to have a pathogenic role in the development of renal injury and further fibrosis (*Nature Reviews Nephrology 2015; 11(2):64*).

This is an interesting work but the Authors should provide a better method and legend description as well as focus the discussion on their findings, and correct the figures as suggested thus providing more evidence of the effect of eMSC.

We rewrote all figure legends in the revised manuscript.

Reviewer #3 (Recommendations for the authors):The abstract starts with the notion that "the immunogenicity and long-term toxicity of artificial carriers hinders the potential clinical translation of this gas therapeutics…" One could argue that, although MSC has been shown to display immunomodulatory effects, MSC from the allogenic origin, as well as the mutant β-galactosidase, may provide novel antigens that can be targeted by the host's immune system. It would be good to discuss this issue.

Thank you very much for your comment. MSCs have long been reported to be hypoimmunogenic or immune privileged; however, several studies have reported the generation of antibodies against MSCs, especially in genetically modified MSCs, such as MSCs transfected with erythropoietin (*Mol. Ther. 17, 369–372 (2009)*). Here, triple reporter genes were applied. In future clinical applications, no therapeutic function genes, Rluc and RFP, should be removed to minimize immune rejection of MSCs.

Previous studies revealed that MSCs do not express blood group antigens or MHC class II antigens (*J Am Soc Nephrol. 25(5), 877-83 (2014)*), and the triple reporter genes, mutant β-galactosidase (β-GAL^H363A^), Rluc and RFP, do not express themselves on the cell membrane, suggesting that there is no rapid immune rejection of genetically modified MSCs. MSCs may exert therapeutic function through a brief "hit-and-run" mechanism, protecting MSCs from immune detection, and prolonging their persistence in vivo may improve clinical outcomes and prevent patient sensitization to donor antigens (*Nature Biotechnology 32, 252–260 (2014)).*It is widely acknowledged that MSCs are highly immunomodulatory (*Cellular and Molecular Immunology 20, 555–557 (2023)*). MSC rejection depends on the balance between MSC expression of immunogenic and immunosuppressive factors (*Nature Biotechnology 32, 252–260 (2014)*)*.* Strategies to improve the persistence of eMSCs should be investigated in future studies.

The authors report in Figure S4 and 1F that the addition of MGP to the MSC elicits a release of NO based on a Griess assay (S4) and staining of DAF-FM. This is a nice result, but it has to be taken into account that Griess and DAF-DA also detect NO indirectly via oxidation products. Therefore, a biological sensor of NO production in the MSC could be the generation of cGMP. cGMP can be easily measured and is, to this reviewer's knowledge, the most likely effector of the effects of NO on MSC. Alternatively, and surprisingly the authors do use a NO analyzer and even EPR! in the in vitro studies, why not use these also for the validation of NO release by the cells in vitro?

Thank you for your comments and advice. As suggested, we conducted additional experiments to validate NO release by eMSCs with MGP administration in vitro using the EPR technique. An obvious characteristic triplet EPR signal (NO signal) was observed in eMSC with MGP administration. The data are shown in Figure 1G.

Figures S6 and S7 demonstrate the modest effect of the MGP-dependent NO generation on H2O2-induced apoptosis and survival. A missing control is the effect of MGP on H2O2-induced apoptosis and survival of the control cells. May spontaneous release of NO, without the help of the galactosidase is enough for the effect?

We have proven by Greiss, EPR and other experiments that MGP does not release NO spontaneously, so the anti-apoptotic effect is mainly due to the synergistic effect of NO release from eMSCs themselves (Figure 5, Figure 5—figure supplement 1-3).

For the in vivo studies described in figure 3 it would be helpful I a description in the methods section would be given on how the MSC are transplanted.

We have added detailed information to the 'Methods' section. In our study, 1×10^6^ eMSCs (30 μL, 3 sites) were transplanted into the left kidney by renal parenchymal injection, and then the muscle layer and skin were successively sutured with 6-0 suture.

The model of IR can be performed in different ways and even though the authors refer to a published paper it should be stated whether this is a unilateral model or a bilateral model and whether or not the contralateral right kidney is dissected (most likely).

We have described more details of the AKI model in the 'Methods' section. This is a unilateral ischemia‒reperfusion injury model with right kidney dissection.

In the angiogenesis studies it is surprising that PBS stimulates angiogenesis, can this be discussed? Again, the MGP-only control is lacking. Nice this is confirmed in HUVEC. Do you have any indication of how long the MSC survives after transplantation?

In Vegfr2-Fluc-KI mice, firefly luciferase (Fluc) was expressed under the promoter of vascular endothelial growth factor receptor 2 (Vegfr2) to report angiogenesis in vivo. Injuries in the sham and PBS groups can also initiate angiogenesis during tissue regeneration, while BLI signals were significantly enhanced in the eMSC with MGP administration group.

We have proven by Greiss, EPR and other experiments that MGP does not release NO spontaneously (Figure 5, Figure 5—figure supplement 1-3), so the anti-apoptotic effect is mainly due to the synergistic effect of NO release from eMSCs themselves.

BLI data (Figure 3A) from our study suggest that by day 7, less than 2% of the transplanted eMSCs are still alive. The pattern of donor cell death in this study is consistent with previous reports.